# Brief communication: A submarine wall protecting the Amundsen Sea intensifies melting of neighboring ice shelves

Özgür Gürses[1], Vanessa Kolatschek[1], Qiang Wang[1], Christian B. Rodehacke[1, 2]

[1]Alfred-Wegener-Institut Helmholtz-Zentrum für Polar- und Meeresforschung, Bremerhaven, D-27570, Germany
[2]Danish Meteorological Institute, Copenhagen Ø, DK-2100, Denmark

*Correspondence to*: Christian B. Rodehacke (christian.rodehacke@awi.de)

**Abstract**

Disintegration of ice shelves in the Amundsen Sea, in front of the West Antarctic Ice Sheet, has the potential to cause sea level rise by inducing an acceleration of ice discharge from upstream grounded ice. Moore et al. (2018) proposed that using a submarine wall to block the penetration of warm water into the sub-surface cavities of these ice shelves could reduce this risk. We use a global sea ice-ocean model to show that a wall shielding the Amundsen Sea below 350 m depth successfully suppresses the inflow of warm water and reduces ice shelf melting. However, these warm water masses get redirected towards neighboring ice shelves, which reduces the net effectiveness of the wall. The ice loss is reduced by 10% integrated over the entire Antarctic continent.

## 1 Introduction

One of the consequences of the warming of Earth's climate is sea level rise (Vaughan et al., 2013). Sea level rise will impact coastal societies, and economic activities in these areas. Currently the main contributors to rising global mean sea level are the thermal expansion of the world's ocean, the mass losses emanating from the Greenland Ice Sheet, and the world-wide recession of mountain glaciers and ice caps (Chen et al., 2017; Rietbroek et al., 2016; Shepherd et al., 2012). The remaining smaller sources are continental ground water depletion (Wada et al., 2012) and the Antarctic Ice Sheet (King et al., 2012; Rietbroek et al., 2016; Shepherd et al., 2012); though Antarctica's sea level contribution has accelerated in recent decades (King et al., 2012; Rietbroek et al., 2016; Rignot et al., 2011).

In Antarctica, remotely sensed, modelled and paleoclimatological-proxy data indicate that the highest potential sea level contribution will come from the West Antarctic Ice Sheet (Bamber et al., 2009; Golledge et al., 2013; Joughin and Alley, 2011; Pollard and DeConto, 2009; Sutter et al., 2016), particularly from the Amundsen Sea Sector, where the progressive thinning of its ice shelves over the last two-and-half decades has greatly enhanced rates of ice mass loss emanating from this sector (Pritchard et al., 2012; Rignot et al., 2014; Shepherd et al., 2018). Here ice shelves currently prevent unrestricted flow of ice streams into the ocean. Here, warm high salinity Circumpolar Deep Water (CDW) has been observed to flow onto the continental shelf and flood the cavities underneath the Amundsen Sea Sector's ice shelves, driving high rates of basal

melting (Depoorter et al., 2013; Jacobs et al., 2011; Jenkins et al., 2018; Pritchard et al., 2012). Various processes control the flow of warm water masses (a body of ocean water with a common formation history and a defined range of tracers, such as temperature and salinity, is called water mass) predominately via glacially scoured submarine troughs (Bingham et al., 2012; Dutrieux et al., 2014) into the ice shelf cavities. It includes wind-driven Ekman transport, whereby variations in offshore wind stresses, also altered by local sea ice conditions (Kim et al., 2017), lift CDW onto the continental shelf (Dutrieux et al.,

2014; Kim et al., 2017; Paolo et al., 2018; Schmidt et al., 2013). During its transport onto the continental shelf the water mass is transformed into modified CDW (mCDW) by mixing with local, fresher on-shelf water masses (Webber et al., 2017, 2018). In the Amundsen Sea, decadal-scale changes in the draft and intensity of the CDW incursion onto the continental shelf – and ultimately the basal melting of the ice masses fringing this sector of the Antarctica – have also been directly linked to changes in the large-scale oceanic and atmospheric circulation, including the influence of ENSO-induced

atmospheric wave trains propagating towards this region from the central tropical Pacific Ocean (Dutrieux et al., 2014; Jenkins et al., 2018; Nakayama et al., 2018; Steig et al., 2012). These processes together drive the detected retreat of ice shelves in the Amundsen Sea through decadal oceanographic variability (Jenkins et al., 2018).

Since the West Antarctic Ice Sheet resides on retrograde sloping bedrock topography (Fretwell et al., 2013), it is inherently susceptible to the Marine Ice Sheet Instability (Schoof, 2007; Weertman, 1974), whereby the reduced buttressing effect of

thinning ice shelves triggers the retreat of upstream ice, leading to larger ice thickness at the groundling line – the groundling line marks the transition from grounded ice to floating ice. This amplifies the ice flux across the new grounding line, which stretches and thins the ice further and ultimately triggers additional grounding line retreat. This sustained retreat accelerates the transport of grounded inland ice towards the ocean past the grounding line, where it directly contributes to sea level rise (by ice berg calving and ocean-driven melting).

Moore et al. (2018) proposed a targeted geoengineering project that could reduce the risk of this ice sheet instability by protecting the ice shelves from warm Circumpolar Deep Water via the erection of a submarine wall. Wolovik and Moore (2018) tested this idea with a simple flow line model (2-dimensional xz-plane model) of Thwaites Glacier – one of the largest contributors of ice discharge into the Amundsen Sea (Shepherd et al., 2018; Turner et al., 2017). In addition to the erection of a submarine wall they (cf. Moore et al., 2018) imposed artificial pinning points to enhance the buttressing effect

of ice shelves on grounded ice. Both measures successfully reduced ice mass loss emanating from this sector of Antarctica (Wolovick and Moore, 2018).

In this paper we investigate how a submarine wall, shielding the Amundsen Sea Embayment (Figure 2a), reduces the basal melting of rates ice shelves flowing into the Amundsen Sea Embayment. The warm water masses rejected by the wall enhance ice shelves west of the wall. These effects counteract the wall's purpose mitigating sea level raise. In this study, we

neglect feedbacks between changes of basal melting rates and advance or retreat, respectively, of impacted ice shelves. We do not analyze how the wall hinders the exchange of nutrients and influences submarine biological processes.

## 2 Model setup

We use a global Finite Element Ocean Model (FESOM; Wang et al., 2014) to test the effects of erecting a wall in front of the Amundsen Sea Sector's ice shelves. The model has a variable horizontal resolution of (minimum 5 km) around Antarctica and its adjacent ice shelf cavities and has 100 vertical levels (z-coordinate). The interaction between the ocean and static ice shelves occurs via the three-equation system that describes the flux of heat and fresh water between the ocean and ice shelf base through an exchange controlling boundary layer (Hellmer and Olbers, 1989; Holland and Jenkins, 1999). FESOM has proven its applicability for oceanographic studies of the Southern Ocean (Hellmer et al., 2012; Nakayama et al., 2014; Timmermann et al., 2012). While coarse resolution models have been found to underestimate the ocean-induced basal melting of Antarctica's ice shelves (Naughten et al., 2018), our basal melting rates (Figure 1b) are in reasonable agreement with recent observational estimates (Rignot et al., 2013). The model utilizes the ocean bathymetry, ice shelf geometry and grounding line position data of RTOPO1 (Timmermann et al., 2010). We use the CORE2 forcing for atmospheric conditions (Large and Yeager, 2008) covering the years 1948—2007 to drive the ocean model. This forcing period is run twice. The first full period is considered as spin-up and, hence, we restrict our analysis on the last complete forcing period.

Considerable oceanic variability has been detected at both seasonal and interannual timescales in front of both Pine Island (Webber et al., 2017) and Dotson Ice Shelf, located between Thwaites Glacier and Getz Ice Shelf, (Jenkins et al., 2018), for instance. It is driven by both local and remote forcing. Hence we shall expect some differences between merged hydrographic observations and a simulated long-term mean, while a reliable climatological data set is lacking for our region of interest. Therefore, we use existing observations for comparison with our simulations under the assumption that available observations represent a quasi-mean state. Measured bottom temperatures, predominantly taken in austral summer by the marine cruises *ANT XI/3* (Miller and Grobe, 1996) and *ANT XXVI/3* (Gohl, 2010), provide confirmation that the simulated bottom temperature distribution is reasonable (Figure 1).

We investigate differences in ice shelf basal melting with (WALL) and without (CTRL) the erection of a wall surrounding the Amundsen Sea (Figure 2a). This feature follows the approximate location of the continental shelf break, and blocks any circulation below 350 m depth, such as the CDW inflow from the deep ocean onto the Amundsen Sea Sector's continental shelf (Figure 2). The wall proposed by Moore et al. (2018) blocks only the channelized flow of warm water in troughs leading directly to Pine Island and Thwaites Glaciers, while our wall with a length of about 800 km, is substantially larger than the originally proposed wall in size, and shields the entire Amundsen Sea Embayment.

## 3 Results

Consistent with oceanographic observations, our CTRL experiment simulates accurately the ingress and delivery of warm mCDW through submarine troughs towards the ice shelves fringing the Amundsen Sea Sector (Figure 1). Measured bottom temperatures, acquired in austral summer, also strongly agree with the spatial distribution of our simulated temperatures, giving confidence in our abilities to accurately predict basal melting in the present study (Figure 1).

Contrary to our CTRL experiment, our erected wall blocks the ocean below 350 m depth and suppresses the direct inflow of CDW to the interior of the Amundsen Sea Embayment in front of the western Marie Byrd Land. Consequently, the simulated ocean is generally cooler (Figure 2a) and fresher within the walled region. This colder water column supports enhanced sea ice formation, which releases brine into the underlying ocean across this region. However, the brine-induced salinification is insufficient to compensate the salinity supply of the unobstructed mCDW inflow.

We also detect a slight cooling of the bottom temperatures east of the walled region. The outflow of cooler water masses from the walled region via the Abbot Ice Shelf's sub-ice shelf cavity (south of Thurston Island) contributes to this cooling (Figure 2a and b). The deflected warm water mass flows westward and rises the temperature on the west side of the walled region. This causes the temperature to rise in the westernmost corner of the walled region around Siple Island, because the warm water mass penetrates via the Getz Ice Shelf (between the groundling line of Antarctica and Siple Island) into the walled region.

In the walled region, the lower ocean temperature reduces melting of ice shelves (Figure 2a and Figure 2c). However, the restrained warm water mass advances into the neighboring region, where ice shelves experience intensified melting and amplified ice mass loss (central and western Getz Ice Shelf; Figure 2b). Therefore, the warm water mass that would have otherwise impacted the Amundsen Sea Embayment, shifts to neighboring ice shelves.

Figure 3 depicts the longitudinal distribution of the simulated basal melting rates around Antarctica, with and without erection of the submarine wall. In the Amundsen Sea Embayment the ice mass loss around Pine Island drops significantly by 85 %. This phenomenon contrasts with increases ice mass loss detected at Getz Ice Shelf (~130° W, eastern Marie Byrd Land), where melting increases by approximately 50%. As discussed above, basal melting is reduced in the western Bellingshausen Sea. In addition to the decreased melting simulated underneath Abbot Ice Shelf, basal melting at George VI Ice Shelf increases by up to 10%. The wall has little impact on basal melting of ice shelves fed by ice streams from the East Antarctic Ice Sheet, with the exception of Amery Ice Shelf, where the rate increases by approximately 5 %. The wall in the Amundsen Sea triggers most probably a perturbation that propagates via the Antarctic Coastal Current towards the Prydz Bay in front of the Amery Ice Shelf. All above reported intensified melting rates are larger than the standard deviation (1-sigma) of the 20 years melting rate.

Beside regional changes of the basal melting rates, we inspect the continental-wide integrated effect. The reduced ice loss in the Amundsen Sea Embayment is larger than the corresponding enhanced melting at the western end of the wall. The total ice loss by ice shelves around Antarctica is 10% lower for the WALL experiment.

## 4 Conclusions

In this study, a submarine wall erected along the continental shelf of the Amundsen Sea is found to suppress the inflow of circumpolar deep water onto the continental shelf. This freshens and cools water masses residing shoreward of the wall, resulting in significantly reduced basal melting rates of the ice-shelves located there. However, inflowing warm Circumpolar

Deep Water (CDW) seaward of this wall is found to be redirected westward towards Getz Ice Shelf, where it enhances basal melting by up to 50%. In particular, the ice shelves to the west (central and west Getz Ice Shelf) show steeply increased melting rates. Hence the wall reduces the ice loss of the most vulnerable ice shelves along the margin of the Western Antarctic Ice Sheet, which is, however, not compensated by enhanced melting in the west. Integrated over Antarctica, the

wall decreases ice loss by 10 %. Our results indicate that suppressing the flow of warm water masses into a restricted group of ice shelves results in redirecting it towards a different location. There it enhances basal melting and, ultimately, amplifies ice mass loss. However, it is an open question if this triggers Marine Ice Sheet Instability in the other shelves, because the stability depends on the distribution of pinning points, sloping of the bed, the depth and width of submarine troughs, and the softness of the bed, for instance. The onshore bed properties of the eastern Marie Byrd Land, where Pine Island and

Thwaites Glaciers are located, are most likely vulnerable to the Marine Ice Sheet Instability. Numerous modelling studies show a relic ice cap in the western Marie Byrd Land on the elevated bed rock topography even after part of the West Antarctic Ice Sheet (WAIS) has collapsed (e.g. DeConto and Pollard, 2016; Feldmann and Levermann, 2015; Golledge et al., 2015; Winkelmann et al., 2015). Hence the western Marie Byrd Land is favorable for a stable situation. Though, the bed properties under the ice are still known insufficiently.

Our fully coupled sea ice-ocean model, which includes ice shelves and ocean-ice shelf interaction, is driven by a prescribed atmospheric forcing. Hence any feedbacks, such as changing ocean surface conditions that impact the atmosphere and change the atmospheric forcing on the ocean, are not included. Therefore, small anomalies between both simulations (CTRL vs WALL), such as those seen in Prydz Bay in front of Amery Ice Shelf or in the George VI Ice Shelf, could vanish if we would include atmosphere-ocean feedbacks. Here only simulations coupled to the atmosphere would allow confirming the

robustness of these features.

The used bedrock topography and ice shelf geometry data set influences the melting rate of individual ice shelf caverns, as it has been shown for the smaller Crosson and Dotson Ice Shelves draining also into the Amundsen Sea (Goldberg et al., 2019). Hence most updated data sets would be preferred; however existing inconsistencies between the most updated data sets, which are seen in differences of the reported groundling positions, free board heights and bedrock elevation, require the

use of a by expert judgement merged data products, such as RTOPO. Therefore, we use RTOPO instead of most updated products. Since our simulations are consistent with former studies using RTOPO, the quality of our simulations could be judged in the light of former studies.

Would we detect the penetration of warm water masses via the Getz Ice Shelf into walled region if we use other bathymetry or bedrock topography data sets? If all ice was grounded between the western end of the wall and the coast line, we would

not see any flow of warm water into the walled region. However we would detect enhanced melting, which may open up a route into the protected region. Hence fully coupled ice sheet-ocean model simulations, where the geometry of ice shelves are changed by melting and refreezing, would reveal the vulnerability of the Getz Ice Shelf. These simulations would also uncover if enhanced melting at the western end of the wall may open a backdoor that open a second route to the ice shelves prone to Marine Ice Sheet Instability.

Regardless of the used bathymetry data set, we are confident that the main findings of this study are robust: a wall shielding the Amundsen Sea Embayment reduces basal melting rates within the protected region, the rejected warm water masses flows along the wall westward, west of the wall warmer water masses drive enhanced basal melting. The wall proposed by Moore et al. (2018), which blocks only the circulation in troughs leading directly to Pine Island and Thwaites Glaciers, would have a length of about 50—100 km and would need 10—50 km$^3$ of material. By comparison, the construction of the

Suez Channel required the excavation of about 1 km$^2$ of material (Moore et al., 2018). The simulated wall (length of about 800 km) is substantially larger than the originally proposed wall in size and it shields the entire Amundsen Sea Embayment. Our results suggest that a too small wall blocking only the water flow in the troughs leading to Pine Island, for instance, might be bypassed by warm water masses. For dynamical reasons the (geostrophic) flow of water masses turns to the left (on the Southern hemisphere), if it is not hindered by a topographic obstacle. Therefore, warm water masses might even

recirculate into the ostensibly protected area if the wall is too small, as the inflow of warm water masses through the Getz Ice Shelf into the walled region suggests. However a small wall that only protects Pine Island successfully, may redirect the warm water to neighboring ice shelves with a retrograde bed (for example Thwaites Glacier). There it increases basal melting and may trigger Marine Ice Sheet Instability. The detected poleward shift of westerly winds in the Southern Ocean under global warming (Miller et al., 2006) may shifts also the coast easterly winds along Antarctica's coast poleward, which

lifts further the interface of warm water masses (isothermal) along the continental slope (Spence et al., 2014). Ultimately warm water masses could enter the continental shelf directly beside the contemporary path following topographic depressions (troughs). Under these circumstances the bypassing of a short wall seems to be inevitable, if the wall does not block the entire Amundsen Sea Embayment from coast to coast.

Iron is a micronutrient essential for algal production in the Amundsen Sea (St-Laurent et al., 2017) and the erected wall

affects its availability. The wall blocks in inflow of warm and iron-rich CDW and influences the outflow of iron-rich glacial melt water coming from melting ice shelves. How the changed nutrient supply impacts the marine biological web or the uptake and sequestration of carbon dioxide by the ocean is unclear and goes beyond this study.

Geoengineering aims to attenuate the impact of the ongoing anthropogenic climate change, such as sea level rise, but the results of this study suggest that such proposals could have adverse side effects. To evaluate these effects of using submarine

walls to protect Antarctica's ice shelves in greater detail, the use of fully coupled ice-sheet-shelf-ocean-atmosphere models should be utilized in future analyses. These models of sufficiently high spatial resolution could simulate accurately changes in sub-ice shelf cavity geometry (including track grounding line migration and ice-shelf thinning) as well as the influx of warm water masses (mCDW) to these locations.

**Acknowledgments**

This work has been financed through the German Federal Ministry of Education and Research (Bundesministerium für Bildung und Forschung: BMBF) project ZUWEISS (grant agreement 01LS1612A). We thank in particular Evan Gowan for

his comments, which greatly improved this manuscript. We thank both anonymous reviewers and Mike Wolovick for their comments and engagement in the review process, which improved the manuscript.

**Author contributions**

CR designed the study and wrote the manuscript. ÖG and QW developed and configured the model. ÖG ran the simulations. ÖG and VK performed the analysis. VK and CR prepared the figures. All authors contributed to the interpretation of the results and proofreading of the manuscript.

**Competing interests**

The authors declare that they have no conflict of interest.

**Code and Data availability**

The FESOM1.4 model code is available at https://swrepo1.awi.de/projects/fesom/ after registration. The here used atmospheric forcing data set named "CORE-II" (Large and Yeager, 2008) is freely accessible online (for example at https://data1.gfdl.noaa.gov/nomads/forms/core/COREv2.html). The topography data set RTOPO could be obtained from https://doi.pangaea.de/10.1594/PANGAEA.741917. The temporal average of the fractional basal melting changes between
the CTRL and the WALL simulations is obtainable from Zenodo via https://dx.doi.org/10.5281/zenodo.3240250. The remaining data is available from the first author ÖG upon reasonable request.

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

**Figures**

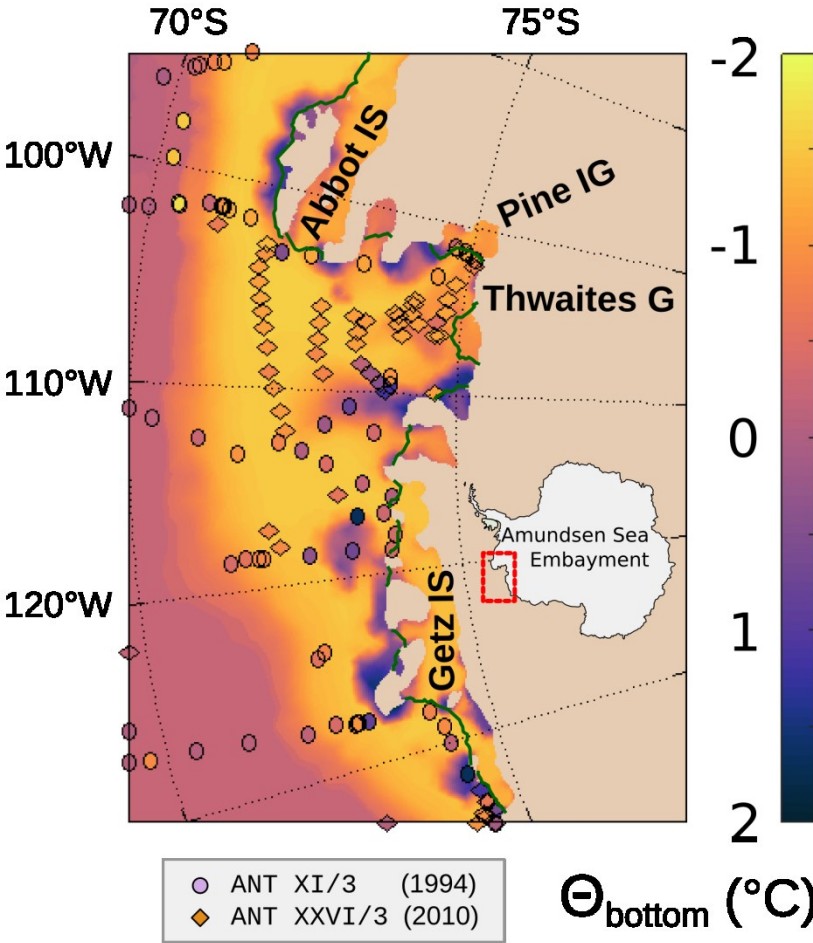


**Figure 1 Modelled and observed seafloor ocean potential temperatures ($\theta_{bottom}$) in the Amundsen Sea Sector of West Antarctica.** Inset shows study location. The plot shows the simulated mean ocean temperatures for the control run (CTRL, years 1948—2007), while individual observed bottom temperatures are represented by circles (ANT XI/3; Miller and Grobe, 1996) or diamonds (ANTXXVI/3; Gohl, 2010) taken in 1994 and 2010, respectively. The shelf ice edge is drawn as solid green line and the inset show
the location of area of interest as red box.

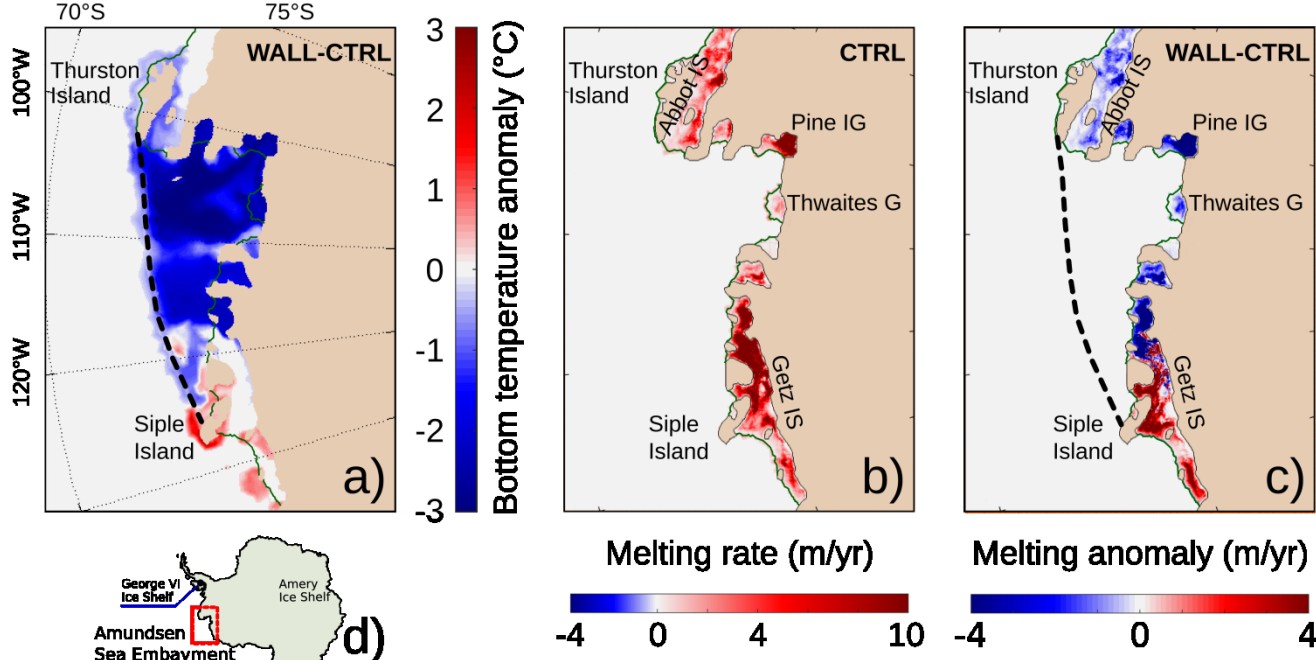

**Figure 2** Simulated potential ocean temperature anomaly (WALL – CTRL) a), simulated basal ice shelf melting rates (CTRL) in b) and its anomaly c). The subplot 2a) shows the simulated potential ocean temperature anomaly (WALL – CTRL) on the seafloor of the Amundsen Sea Embayment and its adjacent ice shelf cavities. The location of the wall is marked as a dashed line and the embayment region is defined in the map d). The middle subplot b) show the simulated melting rates for the control run (CTRL) and the right subplot c) shows basal melting anomaly (WALL - CTRL). The ice shelf edges are highlighted by solid green lines. The following abbreviations are used: Abbot IS (Abbot Ice Shelf), Pine IG (Pine Island Glacier), Thwaites G (Thwaites Glacier) and Getz IS (Getz Ice Shelf).


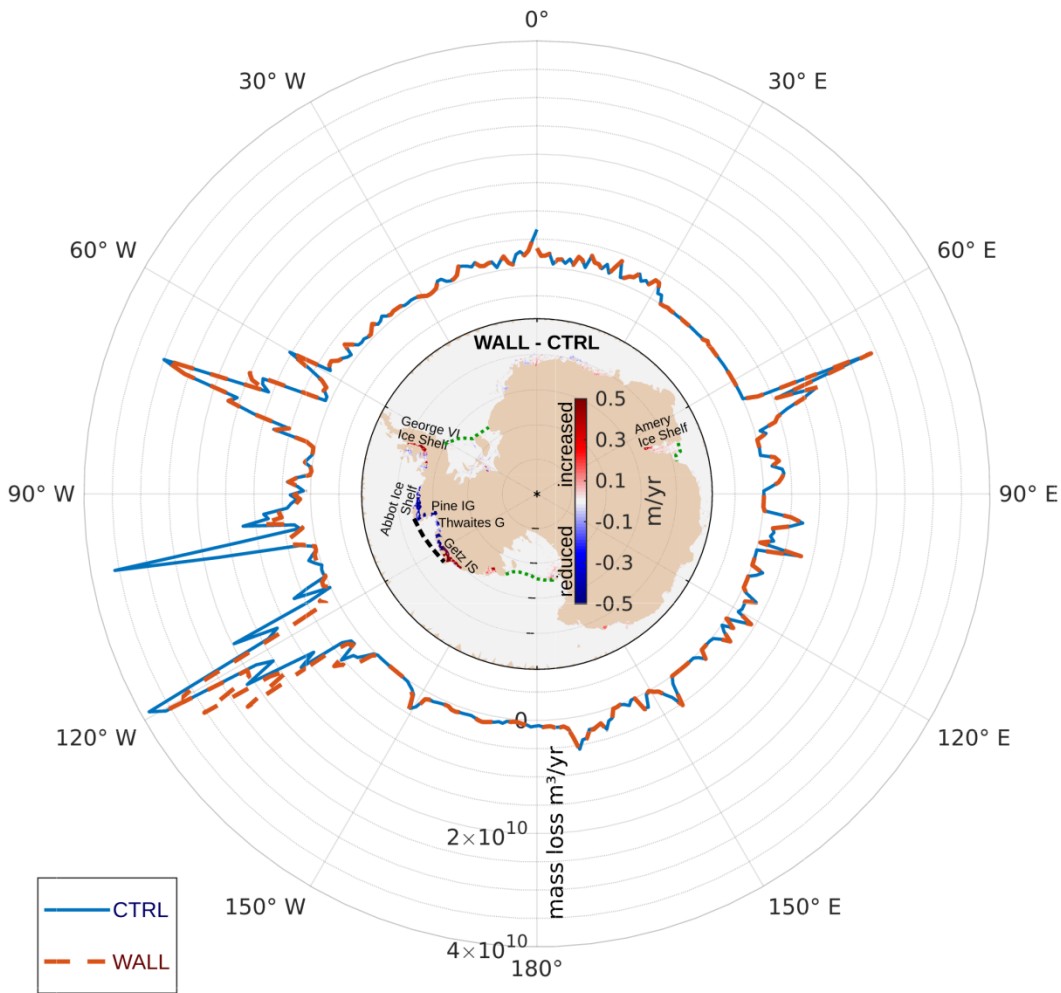

**Figure 3 Mean basal melting rates around Antarctica.** Longitude-specific changes in modelled basal melting with (WALL) and without (CTRL) the presence of the submarine wall are shown as dashed red and solid blue lines surrounding the center map, respectively. The wall's location in the Amundsen Sea is marked by the black dashed line. The map of Antarctica shows the spatial distribution of the melting rate anomaly, where positive numbers (red color) represents increased melting rates if the wall is present (see colorbar). The following abbreviations are used: Pine IG (Pine Island Glacier), Thwaites G (Thwaites Glacier) and Getz IS (Getz Ice Shelf). The shelf ice edges of the Ross Ice Shelf, the Filchner-Ronne Ice Shelf and the Amery Ice Shelf are depicted by green dotted lines.