# Peer review of "Brief communication: A submarine wall protecting the Amundsen Sea intensifies melting of neighboring ice shelves"

_The Cryosphere, 2019_

## Short Comment (SC1) · 29 Mar 2019

Comment on, "Brief Communication:  A submarine wall protecting the Amundsen Sea intensifies melting of neighboring ice shelves", by Gürses et al.

Mike Wolovick

First of all, I would like to congratulate the authors on an excellent modeling paper answering a concise scientific question.  This sort of skeptical engagement from the scientific community is exactly what we were hoping for when we wrote our original geoengineering papers.  It is vitally important that all potential side effects of any geoengineering proposal are explored thoroughly, including side effects that the original authors did not think of.  This side effect certainly falls under that category; we did not anticipate that blocking warm water from reaching some ice shelves would cause it to increase melting at other shelves.

However, I am worried that casual readers might draw the implication that an intervention which merely redirected melting from one ice shelf to another would therefore be ineffective.  It is important to emphasize that what matters from the perspective of human societies is not the floating ice shelves, which already displace their weight in water and thus make no direct contribution to sea level when they melt, but rather the grounded ice, which can raise sea levels if it flows into the ocean.  The floating shelves are important only insofar as they act to buttress the grounded ice and prevent grounding line retreat.  The authors themselves alluded to this issue in the conclusion, writing, "[I]t is an open question if this triggers Marine Ice Sheet Instability in the other shelves, because the stability depends on the distribution of pinning points, sloping of the bed, depth and width of submarine troughs, and the softness of the bed, for instance.  The onshore bed properties of the western Marie Byrd Land, where Pine Island and Thwaites Glaciers are located, are probably more favorable for a stable situation than in the eastern Marie Byrd Land sector."[1]

I would argue that the above statement is far too weak.  We can be highly confident that Pine Island and Thwaites Glaciers are more vulnerable to a runaway Marine Ice Sheet Instability than the areas of western Marie Byrd Land onshore of the Getz Ice Shelf.  At the simplest level, we can look at the basal topography of these areas of the ice sheet (Fretwell et al., 2013).  There is a large overdeepened marine basin onshore of Thwaites Glacier, but there is elevated basal topography, in some places above sea level, onshore of Getz (Fig 1).  Thwaites and Pine Island are also retreating at the present day (Turner et al., 2017), and indeed, there have been reasonable suggestions from both
* * *
1    This statement needs to switch western and eastern, and it is somewhat ambiguously worded at the end.  It would be more accurate to say that the onshore bed properties in eastern Marie Byrd Land, where Pine Island and Thwaites Glaciers are located, are more favorable for a runaway instability than western Marie Byrd Land.

[Figure]

Figure 1: Bed elevation in the Amundsen sector of West Antarctica. Data from BEDMAP2 (Fretwell et al., 2013), visualized as a hillshaded surface with two perpendicular light sources so that all slope orientations are visible. Black lines represent ice front and grounding line. The area of the Getz Ice Shelf where the WALL experiment simulated enhanced basal melt is indicated, as is the elevated basal topography inland of that where the stable relic ice cap forms. This geometry can be compared to the geometry inland of Thwaites, where the bed rapidly deepens and the ice sheet is vulnerable to runaway collapse.

data and models that they have already begun a runaway retreat (Favier et al., 2014; Joughin et al., 2014; Rignot et al., 2014). Furthermore, the geographic position of Thwaites and Pine Island ensures that a runaway retreat there will trigger a general collapse of West Antarctica through a "backdoor" destabilization of the Filchner-Ronne and Ross sectors (Feldmann and Levermann, 2015).

By contrast, ice sheet models almost always show that the ice cap onshore of western Getz is the most stable part of WAIS. The elevated basal topography there (Fig 1) allows a relic ice cap to persist even after the rest of WAIS has collapsed. This relic ice cap can be seen in DeConto and Pollard

[Figure]

Figure 2: Compilation of model ice sheet geometries showing the relic ice cap in western Marie Byrd Land which persists even after the rest of WAIS has collapsed. Original source for each figure indicated. Images represent simple screenshots of the model output as visualized in their respective papers; I did not make any attempt to standardize the displays. My only change was to add a red arrow indicating western Marie Byrd Land in each plot.

(2016); Winkelmann et al. (2015); Golledge et al. (2015); Feldmann and Levermann (2015); and in Pollard and DeConto (2009). I have compiled snapshots of ice sheet geometry from all of those models in Fig 2. The relic ice cap in western Marie Byrd Land is a robust feature of ice sheet models because all of the models are responding to the elevated basal topography in that region. Based on a convergence of evidence from basic MISI theory, observations, and models, we can have a high degree of confidence that Pine Island and especially Thwaites are the most unstable parts of West Antarctica, while the ice cap in western Marie Byrd Land is the most stable part. In fact, that little ice cap is likely

to be the last thing left standing long after the rest of WAIS has collapsed.

The model results presented in this paper indicate that a wall built across the Amundsen Sea Embayment at depth could successfully trade high melt rates at Pine Island and Thwaites for high melt rates at Getz.  The current state of glaciological knowledge strongly indicates that the ice cap onshore of the western Getz Ice Shelf is the most stable part of WAIS, and the overdeepened topography of Pine Island and Thwaites are the most unstable parts[2].  While it is always important to quantify all side effects of a potential geoengineering project, not all ice shelves are created equal in their importance to ice sheet stability and sea level rise.  In my opinion, a geoengineering effort that shifted high melt rates from the most unstable part of WAIS to the most stable part of WAIS would be a smashing success. From the perspective of humanity's interest in a stable sea level, the trade described by this paper is an excellent one.

**References**

DeConto, R. M., & Pollard, D. (2016). Contribution of Antarctica to past and future sea-level rise. *Nature*, *531*(7596), 591–597. https://doi.org/10.1038/nature17145

Favier, L., Durand, G., Cornford, S. L., Gudmundsson, G. H., Gagliardini, O., Gillet-Chaulet, F., et al. (2014). Retreat of Pine Island Glacier controlled by marine ice-sheet instability. *Nature Clim. Change*, *4*(2), 117–121. https://doi.org/10.1038/nclimate2094

Feldmann, J., & Levermann, A. (2015). Collapse of the West Antarctic Ice Sheet after local destabilization of the Amundsen Basin. *Proceedings of the National Academy of Sciences*, *112*(46), 14191. https://doi.org/10.1073/pnas.1512482112

Fretwell, P., Pritchard, H. D., Vaughan, D. G., Bamber, J. L., Barrand, N. E., Bell, R., et al. (2013). Bedmap2: improved ice bed, surface and thickness datasets for Antarctica. *The Cryosphere*, *7*(1), 375 – 393. https://doi.org/10.5194/tc-7-375-2013

Golledge, N. R., Kowalewski, D. E., Naish, T. R., Levy, R. H., Fogwill, C. J., & Gasson, E. G. W. (2015). The multi-millennial Antarctic commitment to future sea-level rise. *Nature*, *526*(7573), 421–425. https://doi.org/10.1038/nature15706

Joughin, I., Smith, B. E., & Medley, B. (2014). Marine Ice Sheet Collapse Potentially Underway for the Thwaites Glacier Basin, West Antarctica. *Science*, *344*(6185), 735–738. https://doi.org/10.1126/science.1249055
* * *
2   Although it is also important to be open to the possibility that our consensus understanding may be wrong.  In particular, I worry that a lack of high density ice thickness measurements in western Marie Byrd Land could be hiding deep subglacial troughs and therefore causing us to overestimate the stability of that region.

Pollard, D., & DeConto, R. M. (2009). Modelling West Antarctic ice sheet growth and collapse through the past five million years. *Nature*, *458*(7236), 329–332. https://doi.org/10.1038/nature07809

Rignot, E., Mouginot, J., Morlighem, M., Seroussi, H., & Scheuchl, B. (2014). Widespread, rapid grounding line retreat of Pine Island, Thwaites, Smith and Kohler glaciers, West Antarctica from 1992 to 2011. *Geophysical Research Letters*, *41*(10), 3502–3509. https://doi.org/10.1002/2014GL060140

Turner, J., Orr, A., Gudmundsson, G. H., Jenkins, A., Bingham, R. G., Hillenbrand, C.-D., & Bracegirdle, T. J. (2017). Atmosphere-ocean-ice interactions in the Amundsen Sea Embayment, West Antarctica. *Reviews of Geophysics*, *55*(1), 235–276. https://doi.org/10.1002/2016RG000532

Winkelmann, R., Levermann, A., Ridgwell, A., & Caldeira, K. (2015). Combustion of available fossil fuel resources sufficient to eliminate the Antarctic Ice Sheet. *Science Advances*, *1*(8), e1500589. https://doi.org/10.1126/sciadv.1500589

---

## Referee Comment (RC1) · Anonymous Referee #1 · 3 Apr 2019

**Review of "Brief Communication: A submarine wall protecting the Amundsen Sea intensifies melting of neighboring ice shelves" by Gürses et al., 2019**

**Summary**

The authors use an ice-ocean model to investigate the effects of a submarine wall on the basal melting of the ice shelves fringing the Amundsen Sea Sector, West Antarctica. While a clear reduction in basal melting shoreward of (and in some cases adjacent to) the wall is detected, an enhanced melting signal is also found along the neighboring Getz Ice Shelf (as well as farther afield at George VI and Amery Ice Shelves), which the authors state may reduce the effectiveness of such a construction. However, despite increased melting across these regions, the large reduction in melting simulated over the Amundsen Sea Sector is believed to contribute to a ~10% decrease in Antarctica's total mass loss. Raising important questions about the usefulness (or otherwise) of geoengineering as a means to mitigate Antarctic ice-mass loss, I therefore believe the findings presented in this manuscript are timely and will be of genuine interest to the readership of The Cryosphere. However, prior to publication, I would encourage the authors to address several important points detailed below.

**General comments**

*Model bathymetry*     In Section 2, the authors detail the construction of the wall in their model, which acts to block the intrusion of circumpolar deep water (CDW) onto the Amundsen Sea's continental shelf. While I unfamiliar with the technicalities of the FESHOM model, I was very surprised to see the use of RTOPO1 in the model setup for bathymetry, ice shelf geometry and grounding line location. This product has now been superseded by at least 3 updated bathymetric models (e.g. Bedmap2 (Fretwell et al., 2013); IBCSO (Arndt et al., 2013); RTOPO2, Schaffer et al., 2016)), which have significantly improved our understanding of the Amundsen Sea Sector's continental shelf and sub-ice shelf cavity geometry via a range of new in-situ observations and model predictions. A simple subtraction of RTOPO1 from IBCSO (Figure 1 of this review) emphasizes this point, and shows substantial between-model differences in bedrock elevation throughout the domain, including underneath the ice shelves.

It is conceivable that these differences may lead to substantial variations in modelled CDW ingress and basal melting throughout the Amundsen Sea Sector, which may in turn have impacts for the corresponding Antarctic-wide melt budgets presented in Figure 3, and potentially the overall conclusions of the paper. In order for the findings of this paper to be convincing, I therefore strongly encourage the authors to rerun their analyses using one or all of these models, and carefully adjust the figures/text as necessary to incorporate any new or additional results.

*Standard of writing/English language*       While I appreciate that English may not be the native language of the authors, I echo the Editor's initial comments that the main text still includes a large amount of verbose and/or non-standard sentence construction, which at times makes the flow of the manuscript difficult to follow and/or comprehend. This is particularly true of the end of Sections 3 and 4, where the authors concluding statements appear to downplay the importance of intensified neighboring melt - the focus of the title and abstract (see specific comments below), and thus what I initially perceived to be the key message of this research. I have attempted to restructure large parts of the main text to the best of my ability, but prior to publication I would again ask the authors to very carefully read through their manuscript with the assistance of a native English speaker/proofreader, to improve the readability of this otherwise interesting piece of research.

*Citations*        Whilst the style of referencing in this manuscript is generally satisfactory, I think the main text is somewhat marred by an over-reliance of modelling-based studies, and omits

a lot of other key research on (e.g. observationally constrained) Amundsen Sector ice-ocean-atmosphere interactions and/or glacial change. Such citations should be added to the text to provide a more reasoned/well-rounded discussion. Occasionally, citations are also omitted from sentences altogether, which should also be addressed. (See my suggested edits in the specific comments below).

*Introduction*    At the end of the introduction section, I think some words on the flaws and critical 'next steps' of the studies presented by Moore et al. (2018) and Wolovik and Moore (2018) should be added, to qualify the present study and emphasize to the reader why modelling the impacts of building such a wall might be required. The inclusion of a sentence similar to the one on Lines 116-117 could also be added to contextualize the wider role of geoengineering, and hence the need to accurately predict 'adverse side effects'.

*Section 3 (Lines 63-64)*        Following Section 2 (Lines 56-57), are your modelled 1947-2007 ocean temperatures also restricted to summertime means? Or do they reflect annual averages? I think this might be worth explicitly stating here. Similarly, if indeed they do reflect annual averages, then have you also considered the importance of seasonal changes in CDW ingress onto the continental shelf, as has been noted in the recent literature? (e.g. Thoma et al., 2008; Steig et al., 2012; Dutrieux et al., 2014; Webber et al., 2017). Such changes may lead to large variations in bottom temperatures over seasonal timescales (and hence basal melt rates), which may not be representative of the *in-situ* temperatures shown in Figure 1 of the manuscript. If this is the case, then what steps have been taken to validate the temperatures estimated by your model during non-summer seasons?

**Specific scientific comments**

Ln 74 – "*The warm water mass penetrates through the Getz Ice Shelf into the walled region*". Following my concerns on the use of RTOPO1 above, is this phenomenon present when the model is run with more updated cavity geometry information (e.g. IBCSO/RTOPO2)? Equally, what impact does this have on the simulated spatial distribution and magnitude of melting of Abbot Ice Shelf? In Figure 1 of this review, it is apparent that significant (> +/- 250 m) differences exist underneath these ice shelves, so I would encourage the authors to give this careful consideration.

Lns 85 to 87 – These sentences appear highly speculative and in physical terms, I don't understand how this could be the case. The positioning of the ACC over the Bellingshausen Sectors' continental shelf break has been implicated as the predominant driver of unmodified CDW flooding across this region (e.g. Holland et al., 2010; Bingham et al., 2012; Schmidtko et al., 2014; Wouters et al., 2015; Paolo et al., 2015; Christie et al., 2016; Zhang et al., 2016; Hogg et al., 2017), which is presumably the overriding driver of melt variability at GVIIS. As such, I don't understand how mCDW, which would presumably be constantly freshening during its transport underneath and eastward of the Abbot Ice Shelf, could either reach GVIIS or play a more important role than the influence of the ACC here. I would encourage the authors to carefully consider this point and either clarify why they think this to be the case, and/or amend the text/interpretations as necessary.

The same comment applies to why they think reductions in melt rate in the Amundsen Sector may influence melting at Amery Ice Shelf. Presumably any propagation in the coastal current would become entrained within the Ross Gyre, and not extend to the other side of the continent (cf. Nakayama et al., 2014; Dotto et al., 2018)? Assuming it did, however, then presumably any diverted CDW would again be freshened during its advection towards these regions? As above, I'd like to see a more convincing discussion of why the authors believe this to be the case added here.

I am also interested to see how these findings may change when the model is forced with more updated bathymetry as discussed above. While Figure 1 in this review only shows the Amundsen Sea Sector and its surrounds, significant differences in bathymetry also exist around the continent.

**Technical comments**

Title – For those unfamiliar with the geography of Antarctica, I would reword the title to "*A submarine wall protecting the Amundsen Sea, West Antarctica, intensifies melting of neighboring ice shelves*" or similar.

Ln 8 – Add "*Sector of West Antarctica*" after '*Amundsen Sea*'. Also reword the end of the sentence to "…*acceleration of ice discharge from upstream grounded ice*" for technical accuracy.

Ln 9 – '*et al*' is a Latin abbreviation for '*et alia*', and so a period should follow the '*al*' (i.e. '*et al.*'). I have noticed this small error throughout the manuscript, so the authors should address this universally throughout the document. Also, add the word '*ocean*' between '*warm water*'.

Ln 10 – Suggest rephrasing the end of this sentence to "…*into the sub-surface cavities of these ice shelves could reduce this risk*". The word '*sea*' preceding '*ice-ocean*' model is not needed, and should be removed.

Ln 11 – Change '*warm water*' to '*this water*'. Rephrase next sentence to begin "*However, these water masses get redirected … which reduces the net effectiveness …*".

Ln 14 – Should read "… *the warming of Earth's climate is sea level rise*". Add a reference to the IPCC (e.g. Vaughan et al., 2013) to the end of the next sentence.

Ln 15 – Suggest rewording to "*Currently, the main … mean sea levels are the thermal expansion of the world's oceans, the mass losses emanating from the Greenland Ice Sheet, and the world-wide recession of mountain glaciers and ice caps…*".

Ln 17 – Suggest rewording to "… *and the ice mass losses originating from the Antarctic Ice Sheet… although Antarctica's…*". (Note here the capitalization of the pronoun 'Antarctic Ice Sheet'). At the end of this sentence, a reference to Shepherd et al. (2018) should also be added.

Ln 20 – Suggest rewording this sentence to read "*In Antarctica, remotely sensed, modelled and palaeoclimatological-proxy data indicate that the highest potential for sea level rise will come from the West Antarctic Ice Sheet (Joughin and Alley, 2011), particularly from the Amundsen Sea Sector, where the progressive thinning of its ice shelves over the past ~25 years has greatly enhanced rates of ice mass loss emanating from this sector*" or similar. At the end of this sentence, cite e.g. Pritchard et al. (2012); Mouginot et al. (2014); Rignot at al. (2014); Paolo et al., 2015; Shepherd et al. (2018).

Ln 22 – Suggest rewording next sentence to something like: "*Here, warm, high salinity circumpolar deep water (hereafter CDW) has been observed to flow onto the continental shelf and flood the cavities underneath the Amundsen Sea Sector's ice shelves, driving high rates of basal melting*". Add citations (e.g. Jenkins et al., 2010; Pritchard et al., 2012; Rignot et al., 2013; Jacobs et al., 2013; Depoorter et al., 2013) here.

Lns 25-26 – Merge these two sentences for brevity. Could read something similar to: "*Various processes… ice shelf cavities, including, most predominantly, wind-driven changes in Ekman transport, whereby variations in offshore wind stresses lift CDW onto the continental shelf*". An abundance of new literature has been published on this phenomenon in recent years,

which could/should be cited here in addition to work by Kim et al (2017). These include, but are not limited to: Thoma et al. (2008); Steig et al. (2012); Jacobs et al. (2013); Dutrieux et al. (2014); Walker et al. (2017); Christie et al. (2018); Greene et al. (2018) and Paolo et al. (2018).

Ln 27 – Suggest rewrite to: "*During its transport onto the continental shelf, this water mass is … by mixing with local, fresher on-shelf water masses*". A citation is also needed here (suggest Webber et al. (2017)).

Lns 25-29 – Somewhere in this section I think a short sentence should be added detailing the important role submarine troughs play in amplifying the transmission of CDW to the grounding line (following e.g. Nitsche et al. (2007); Bingham et al. (2012); Dutrieux et al. (2014)). The addition of this sentence would critically also give context to the discussion presented in Section 3 (Line 62).

Ln 26 – Suggest reworking the rest of this paragraph to the following or similar for conciseness: "*In the Amundsen Sea Sector, decadal-scale changes in the draft and intensity of CDW incursion onto the continental shelf – and ultimately the basal melting of the ice masses fringing this sector of Antarctica - have also been directly linked to changes in global-scale atmospheric circulation, including the influence of ENSO-induced atmospheric wave trains propagating towards this region from the central tropical Pacific Ocean (Steig et al., 2012; Dutrieux et al., 2014; Jenkins et al., 2018; Nakayama et al., 2018; Paolo et al., 2018)*".

Ln 32 – Suggest the amalgamation of this and the following sentence for conciseness. Could read something like: "*Since the West Antarctic Ice Sheet resides on retrograde sloping topography (Mercer, 1978), it is inherently susceptible to a Marine Ice Sheet Instability, whereby the reduced buttressing effect of thinning ice shelves triggers the retreat of upstream ice, leading to larger ice thicknesses at the grounding line (Hughes, 1973; Weertman, 1974; Schoof, 2007)*". [Note also here the addition of several classic papers I was surprised to not see in the text. Also, as the term 'grounding line' hasn't been introduced, I would consider also defining this in a short, follow-up sentence].

Ln 35 – Hyphen required between '*grounding line*'. For clarity, next sentence could also be amended to read: "*This sustained retreat accelerates the transport of inland ice towards the ocean past the grounding line, where it directly contributes to sea level rise*".

Ln 38 – Full stop required after the abbreviation '*al*' as discussed above. Also, suggest changing '*this ice sheet collapse mechanism*' to '*marine ice sheet instability*' since this has just been defined above.

Ln 39 – Suggest changing '*warm water with*' to '*CDW via the erection of*'.

Ln 40 – '*Thwaites Glacier*' is a pronoun, hence the word '*the*' directly preceding it should be omitted. Also suggest reword of the end of this sentence to "*…Thwaites Glacier – one of the largest contributors of ice discharge into the Amundsen Sea (Rignot et al., 2011; Mouginot et al., 2014; Turner et al., 2017; Shepherd et al., 2018)*" for clarity. [Note the addition of several key recent citations here].

Ln 41 – This sentence is highly repetitive of the preceding sentence explaining the work of Moore et al. (2018), but can easily be fixed by changing to something like: "*In addition to the erection of subsurface walls (cf. Moore et al., 2018), they imposed artificial pinning points to enhance the buttressing effect of ice shelves on grounded ice. Both measures were found to successfully reduce ice mass losses emanating from this sector of Antarctica*".

Ln 42 – As noted in my general comments, some words on what these studies didn't examine/consider (i.e. the potentially adverse effects elsewhere), in order to qualify the research presented in this paper, should be added here.

Ln 45 – Should read "*Amundsen Sea Sector's ice shelves*". Next sentence should also read "*…horizontal resolution (minimum 5km) around Antarctica and its … and has 100 vertical levels (z-coordinate).*" for clarity.

Ln 49 – Should references be listed in chronological order? Also suggest rewording following sentence to "*While coarse resolution ocean models have been found to underestimate the ocean-induced melting of Antarctica's ice shelves, our basal melting rates are in reasonable agreement with recent observational estimates*". [The authors should also add appropriate citations to the observational estimates they refer to, as well as a cross reference to their Figure 2b here].

Ln 52 – Suggest using the word '*of*' in place of *'from'* for grammatical accuracy. See also my comments above regarding my concerns over the use of RTOPO1.

Ln 54 – Suggest change to "*This forcing period is run twice*".

Lns 58-60 – I think these two sentences could be reworked to become much easier to read/comprehend. Suggest reword to: "*We investigate differences in ice shelf basal melting with (WALL) and without (CTRL) the erection of a wall surrounding the Amundsen Sea (Figure 2a)*" [see also my comments on the manuscript's figures below]. Then: "*This feature follows the approximate location of the continental shelf break (~1000 m), and blocks CDW inflow from the deep ocean onto the Amundsen Sea Sector's continental shelf*".

Ln 62 – Suggest amalgamating the first two sentences for clarity and conciseness. "*Consistent with oceanographic observations [Authors should add reference to the appropriate citations and/or manuscript figure here], our CTRL experiment simulates accurately the ingress and delivery of mCDW through submarine troughs towards the ice shelves fringing the Amundsen Sea Sector*". [Note also that the place name 'Amundsen Sea Embayment' is used here for the first time. This has not been introduced prior to this line, so I would suggest using either 'Amundsen Sea Sector' or 'Amundsen Sea Embayment' universally throughout the manuscript for consistency].

Ln 63 – Suggest '*acquired'* in pace of '*taken'*. I would also consider rephrasing this sentence for clarity to "*… acquired in austral* summer *(cf. Section 2), also strongly agree with the spatial distribution of our simulated temperatures, giving confidence in our abilities to accurately predict basal melting in the present study*" or similar.

Ln 65 – This sentence is highly verbose, and could be shortened considerably. Suggest something like: "*Contrary to our CTRL experiment, our erected wall blocks the ocean below 350 m depth and suppresses the direct inflow of CDW to the interior of the Amundsen Sea*".

Ln 67 – Change *'(Figure 2)'* to *'(Figure 2 a)'* for clarity of reading/reference to figures [see also my comments on the manuscript's figures below]. I also suggest restructuring the following sentence to "*Enhanced sea ice formation is also simulated, enabled by a resulting colder water column and the consequent release of brine into the underlying ocean across this region*".

Ln 68 – I found the context of this sentence almost impossible to comprehend without reading the next paragraph, so I'd suggest rewording to the following, and also inserting a cross reference to Figure 2. Sentence could read something like: "*However, despite the brine-induced salinification of the water column here, this phenomenon is insufficient to maintain the pronounced melt rates observed in the presence of unobstructed mCDW inflow (cf. Figure 2),*

*as discussed below*". [NB.: brine is by definition salty, hence the inclusion of the word 'salty' is superfluous].

Ln 70 – The construction of this sentence is again rather difficult to comprehend, and can be simplified by saying something like: *"…, which lies shoreward of the easterly Antarctic Coastal Current residing over the continental shelf break at this location*". [Note: A citation should also be added here].

Ln 71 – Suggest changing the word '*through'* with '*via'*.

Ln 72 – Suggesting rephrasing part of this sentence to "*the Abbot Ice Shelf's sub-ice shelf cavity (south of Thurston Island) contributes to this cooling (Figures 2a and b)*". [Note also the added cross reference to Figures 2a and b].

Ln 72 (sentence beginning "*The deflected …*") – Suggest changing the beginning of this sentence to "*Seaward of this wall, mCDW …*", and amalgamating this and the next sentence together. (At present, they are highly repetitive, and could easily be reformulated into one concise statement).

Ln 76 – Add reference to your Figures 2b and c. In the next sentence, add a comma after '*However'*.

Ln 77 – For ease of reading/cross reference to your Figure 2, I would suggest changing the contents of the parentheses to "(*central and western Getz Ice Shelf; Figure 2c*)".

Ln 78 – Add a comma after the word '*therefore'*, remove the comma after '*mass'*, and add the word '*have'* prior to '*impacted'*. Also suggest changing the word '*fringing'* to '*neighboring'* in line with the manuscript's title.

Ln 80 – "*longitudinal dependence*". I'm not sure this is the correct term, given that longitude itself does not directly contribute to the basal melting of ice. '*Longitudinal distribution*' would perhaps be more suitable. Also, at the end of this sentence, I suggest the authors add "… *Antarctica, with and without the erection of the submarine wall*" for clarity.

81 – Embayment or Sector? See my comment re: Ln 62. Also suggest merging the end of this and the next sentence to: "*In the Amundsen Sea Sector [Embayment?], ice mass losses around Pine Island Glacier drop by 85%. This phenomenon contrasts with the increased ice mass loss observed at Getz Ice Shelf as discussed above (see also Figure 2c), where melting increased by ~50%.*".

Ln 83 (sentence beginning "*In the western Bellingshausen Sea*") – This sentence is highly repetitive of the content discussed in Lines 70-74, so could easily be removed or integrated with Lines 70-74.

Ln 85 – Suggest rewording this sentence to "*In addition to the decreased melting simulated underneath Abbot Ice Shelf, basal melting at George VI Ice Shelf increased by up to 10%.*". [Note also that the GVIIS resides on the western flank of the Antarctic Peninsula, not west of the Peninsula].

Ln 87 – Add a comma after '*East Antarctic Ice Sheet'*.

Ln 90 – Following my general comment above, the concluding remarks of this sentence are hard to comprehend, and appear to underplay the key message of the title and abstract. Do you mean to say that while localized melting is enhanced across some neighboring ice shelves, these signals are minimal compared with the simulated continent-wide reductions in melt elsewhere? If this the answer to my question is yes, which I suspect to be the case, then

I'd recommend amending the title, abstract and conclusions to provide a more focused argument in favor of this point. In any case, some rephrasing of this sentence is needed to make your conclusions explicitly clear.

Ln 94 – Suggest beginning with "*In this study, a submarine wall erected along the continental shelf of the Amundsen Sea is found to suppress the inflow of circumpolar deep water onto the continental shelf. This freshens water masses residing shoreward of the wall, resulting in significantly reduced basal melting rates of the ice-shelves located there. However, inflowing CDW seaward of this wall is found to be redirected westward towards Getz Ice Shelf, where it enhances basal melting by up to 50%...*".

Lns 98-101: Like the concluding remarks of Section 3, it is difficult to understand with absolute certainty what the key take home message is from these sentences. Is it the fact that the melting enhances in neighboring regions as a result of constructing a wall, or that these enhanced melting signals are minimal when compared to the Antarctica's overall mass budget? The authors should rephrase this section to make this explicitly clear. Also, given the opening sentences of the conclusion, there is a lot of redundancy/repetition on how CDW is diverted to Getz and causes enhanced losses in this section, which should be removed.

Lns 101-105 – This section comprises mainly of MISI theory, which was covered in the introduction, and so is not required here. I'd recommend removing this entire section, and instead give brief mention to MISI in the following section (see comment below). On a side note, while I suggest this part of the discussion be excised from the text, I also completely disagree that Thwaites and Pine Island Glaciers have the potential to be more stable than the Marie Byrd Land Sector, owing to the deeply bedded, retrograde bed slopes and subglacial basins they reside on (e.g. Bedmap2, RTOPO1, RTOPO2, IBCSO, ALMAP etc.). Also, I presume this sentence contains a typo in that '*eastern Marie Byrd Land Sector*' should actually read '*western Marie Byrd Land Sector*' (i.e. the region flowing into Getz Ice Shelf)?

Lns 106-115 – The construction of this paragraph is very hard to follow and should be edited to offer a more fluid and concise discussion. I suggest the following rewrite, in this particular order:

1) A very brief summary of what building a wall means in terms of basal melting in the Amundsen Sea Sector (including Getz);
2) How the findings of this research compare to the ideas presented by Moore et al. (2018), and what the implications of building the shorter wall he discusses would likely be on this region, and then;
3) What the implications of both walls would therefore be in terms of MISI, and Antarctica's future contributions to sea level rise.

Ln 116 – In light this paper's findings, I recommend editing the end of this sentence to read "*…, but the results of this study suggest that such proposals could have adverse side effects*". Then begin the next sentence with something like: "*To evaluate the effects of using submarine walls to protect Antarctica's ice shelves in greater detail, the use of fully coupled ice-sheet-shelf-ocean models should be utilized in future analyses. These models should be of sufficiently high resolution to simulate accurately changes in sub-ice shelf cavity geometry (including grounding-line migration and ice-shelf thinning), as well as the influx of mCDW to these locations*".

Ln 121 – Suggest removing this sentence, as all it serves to do is cast doubt on the validity of the findings presented in this paper!

Ln 126 – Should read "*… for his comments, which greatly improved this manuscript*".

Ln 129 – Should read "*contributed to the interpretation of the results and proofreading of the manuscript*".

Ln 137 – The full stop after '*Germany*' is not needed here.

Ln 186 – '*Cryopsh.*' Should be changed to '*Cryosphere*'.

Lns 187-235 – Remove.

Figure 1: comments on Figure – I would suggest rescaling this image (particularly all lon/lat labels and color bar size) to more closely align with the scaling of Figures 2 and 3, as its current scaling looks rather odd in comparison. To assist the reader, it would also be highly beneficial to add the ice shelf limits as thin lines onto this plot, similar to those presented in Figure 2. Being picky, I also dislike the sizing and positioning of the glacier and ice shelf labels, which could easily be resized/positioned to be more aesthetically pleasing. If possible, I'd also suggest rotating the figure 90 degrees to align with the orientation of the polar stereographic plots shown in Figures 2 and 3.

Figure 1: comments on caption – For overall clarity and conciseness, I would suggest rewriting parts of the caption as follows: "*Figure 1 – Modelled and observed seafloor ocean potential temperatures in the Amundsen Sea Sector of West Antarctica. Inset shows study location. The plot shows … acquired in 1994 and 2010, respectively*".

Figure 2: comments on figure –

- Each sub-plot should be labelled (e.g. a, b, c) to assist the readability of the text. These changes should then be incorporated into the main text and figure caption as necessary.
- I would also add ice shelf outlines to the left panel as their current omission looks odd.
- I would like to see ice shelf limits also added to the inset map for wider geographical context.
- Why is the wall shown in some plots but not others? Suggest adding it to all plots. For consistency, I also suggest using the same color of dashed line in all plot.
- Why does the spatial extent of the wall change between figures? Please show the exact location of the wall as defined in your model in all plots.
- While the arrangement of the figure is generally satisfactory as is, could the right-hand panels be made bigger (at the slight expense of the left-hand panel's size) by arranging all figures side-by-side in a 1 row x 3 columns fashion? At present, it is quite difficult to see the interesting spatial details contained in the melt maps, which may be remedied by making these figures larger.
- Relatedly, I find the ice front positions in the right-hand panels almost impossible to see against the blue color scale, which would be improved by enlarging the plots. Also, I'd suggest making them thicker and/or a different color (e.g. black) to make them easier to visualize.
- The label for Abbot IS goes off the plot and looks ugly. Suggest writing over 2 lines to neaten this up.

Figure 2: comments on caption – Unlike Figures 1 and 3, the caption of this plot is missing a short opening summary of what the figure shows, which should be added for consistency. Ln 246 – Using my labelling convection, I'd suggest editing this sentence to read "*Figure 2a shows simulated ocean potential temperature anomalies (WALL-CTRL) on the seafloor of the Amundsen Sea and its adjacent ice shelf cavities. The location of the wall is denoted by a dashed line….*". Ln 250 – A colon should follow the word 'used' (i.e. "*The following abbreviations are used: …*"). Ln 252 – Suggest shortening the last sentence to "*Inset shows*

*study location and other regions referred to the text*".  Change all instances of e.g. '*left subplot shows*' to new, explicitly labelled equivalents here and in the main text.

Figure 3: comments on figure –

- Why is color scale inverted in this plot relative to Figure 2? This is extremely confusing for the reader, and should be amended. To add to this confusion, the labels associated with the color bar appear to be incorrect, whereby, according to the current caption, red should actually denote "*shrink*".
- Suggest changing '*shrink'* and '*gain'* to '*decreased'* and '*increased'* melt, respectively.
- Like the right-hand plots in Figure 2, ice shelf outlines should be added to this figure.
- It's very hard to see the spatial detail of melting around Antarctica in the current figure, which is a shame, so I'd also strongly suggest increasing the scale of the center map if possible, or including the addition of inset subplots zoomed over key areas (e.g. GVIIS and Amery Ice Shelf) if not.
- Similarly, given the subtle changes in melting simulated underneath Amery Ice Shelf, it would be helpful to provide a zoom-in inset of the CRTL vs. WALL signals shown in the figure for this region.

Figure 2: comments on caption –Ln 255 – '*Outer ring*' is confusing, so I'd suggest rewording to: "*Longitude-specific changes in modelled basal melting with (WALL) and without (CTRL) the presence of the submarine wall are shown as dashed red and solid blue lines surrounding the center map, respectively*". Ln 257 – "*in the center map*" is superfluous, and should be removed (it is obvious where the black dashed line is).

[Figure]

**Figure 1** – Difference between IBCSO and RTOPO1 seafloor bathymetry (red, IBCSO is deeper; blue, shallower). How do these differences (and/or those of e.g. RTOPO2) affect your modelled changes in CDW incursion/basal melting within a) the Amundsen Sea Sector and b) the rest of Antarctica following the erection of the wall?

**References**

Arndt, J. E., H. W., Schenke, M. Jakobsson, F. O. Nitsche, G. Buys, B. Goleby, M. Rebesco, F. Bohoyo, J. Hong, J. Black, R. Greku, G. Udintsev, F. Barrios, W. Reynoso-Peralta, M. Taisei, and R. Wigley (2013), The International Bathymetric Chart of the Southern Ocean (IBCSO) Version 1.0—A new bathymetric compilation covering circum-Antarctic waters, *Geophysical Research Letters*, 40, 3111-3117, doi:10.1002/grl.50413.

Bingham, R. G., F. Ferraccioli, E. C. King, R. D. Larter, H. D. Pritchard, A. M. Smith, and D. G. Vaughan (2012), Inland thinning of West Antarctic ice-sheet steered along subglacial rifts, *Nature*, 487, 468–471, doi:10.1038/nature11292.

Christie, F. D. W., R. G. Bingham, N. Gourmelen, E. J. Steig, R. R. Bisset, H. D. Pritchard, K. Snow, and S. F. B. Tett (2018a), Glacier change along West Antarctica's Marie Byrd Land Sector and links to inter-decadal atmosphere–ocean variability, *The Cryosphere*, 12, 2461-2479, doi:10.5194/tc-12-2461-2018.

Christie, F. D. W., R. G. Bingham, N. Gourmelen, S. F. B. Tett, and A. Muto (2016), Four-decade record of pervasive grounding line retreat along the Bellingshausen margin of West Antarctica, *Geophysical Research Letters*, 43, 5741–5749, doi:10.1002/2016GL068972, 2016.

Depoorter, M. A., J. L., Bamber, J. A. Griggs, J. T. M. Lenaerts, S. R. M. Ligtenberg, M. R., van den Broeke, and G. Moholdt (2013), Calving fluxes and basal melt rates of Antarctic ice shelves, *Nature*, 502, 89-92, doi:10.1038/nature12567.

Dotto, T. S., A. N. Garabato, S. Bacon, M. Tsamados, P. R. Holland, J. Hooley, E. Frajka-Williams, A. Ridout, M. P. and Meredith (2018), Variability of the Ross Gyre, Southern Ocean: drivers and responses revealed by satellite altimetry, *Geophysical Research Letters*, 45(12), 6195-6204, doi:10.1029/2018GL078607.

Dutrieux, P., J. De Rydt, A. Jenkins, P. R. Holland, H. K. Ha, S. H. Lee, E. J. Steig, Q. Ding, E. P. Abrahamsen, and M. Schröder (2014), Strong Sensitivity of Pine Island Ice-Shelf Melting to Climatic Variability, *Science*, 343, 174-178, doi:10.1126/science.1244341.

Fretwell, P., et al. (2013), Bedmap2: Improved ice bed, surface and thickness datasets for Antarctica, *The Cryosphere*, 7, 375–393, doi:10.5194/tc-7-375-2013.

Greene, C. A., D. D. Blankenship, D. E. Gwyther, A. Silvano, and E. van Wijk (2017), Wind causes Totten Ice Shelf melt and acceleration, *Science Advances*, 3(11), e1701681, doi:10.1126/sciadv.1701681.

Hogg, A. E., A. Shepherd, S. L. Cornford, K. H. Briggs, N. Gourmelen, J. A. Graham, I. Joughin, J. Mouginot, T. Nagler, A. J. Payne, E. Rignot, and J. Wuite (2017), Increased ice flow in Western Palmer Land linked to ocean melting, *Geophysical Research Letters*, 44(9), 4159–4167, doi:10.1002/2016GL072110.

Hughes, T. (1973), Is the West Antarctic Ice Sheet Disintegrating?, *Journal of Geophysical Research*, 78(33), 7884-7910, doi:10.1029/JC078i033p07884.

Holland, P. R., A. Jenkins, and D. M. Holland (2010), Ice and ocean processes in the Bellingshausen Sea, Antarctica, *Journal of Geophysical Research*, 115, C05020, doi:10.1029/2008JC005219.

Holt, T. O., H. A. Fricker, N. F. Glasser, O. King, A. Luckman, L. Padman, D. J. Quincey, M. R. and Siegfried (2014), The structural and dynamic responses of Stange Ice Shelf to recent environmental change, *Antarctic Science*, 26, 646–660, doi:10.1017/S095410201400039X.

Jacobs, S., C. Giulivi, P. Dutrieux, E. Rignot, F. Nitsche, and J. Mouginot (2013), Getz Ice Shelf melting response to changes in ocean forcing, *Journal of Geophysical Research: Oceans*, 118,1–17, doi:10.1002/jgrc.20298.

Jenkins, A., P. Dutrieux, S. S. Jacobs, S. D. McPhail, J. R. Perrett, A. T. Webb, and D. White (2010), Observations beneath Pine Island Glacier in West Antarctica and implications for its retreat, *Nature Geoscience*, 3(7), 468–472, doi:10.1038/ngeo890.

Jenkins, A., D. Shoosmith, P. Dutrieux, S. Jacobs, T. W. Kim, S. H. Lee, H. K. Ha, and S. Stammerjohn (2018), West Antarctic Ice Sheet retreat in the Amundsen Sea driven by decadal oceanic variability, *Nature Geoscience*, 11, 733-738, doi:10.1038/s41561-018-0207-4.

Mercer, J. (1978), West Antarctic ice sheet and CO2 greenhouse: a threat of disaster, *Nature*, 271, 321-325.

Mouginot, J., E. Rignot, and B. Scheuchl (2014), Sustained increase in ice discharge from the Amundsen Sea Embayment, West Antarctica, from 1973 to 2013, *Geophysical Research Letters*, 41, 1576-1584, doi:10.1002/2013GL059069.

Nakayama, Y., R. Timmermann, C. B. Rodehacke, M. Schroder, and H. H. Hellmer (2014), Modeling the spreading of glacial meltwater from the Amundsen and Bellingshausen Seas, *Geophysical Research Letters*, 41, 7942–7949, doi:10.1002/2014GL061600.

Nitsche, F. O., S. S. Jacobs, R. D. Larter, and K. Gohl (2007), Bathymetry of the Amundsen Sea continental shelf: Implications for geology, oceanography, and glaciology, *G³ Geochemistry, Geophysics, Geosystems*, 8(10), doi:10.1029/2007GC001694.

Paolo, F. S., H. A. Fricker, and L. Padman (2015), Volume loss from Antarctic ice shelves is accelerating, *Science*, 348, 327–331, doi:10.1126/science.aaa0940.

Paolo, F. S., L. Padman, H. A. Fricker, S. Adusumilli, S. Howard, and M. R. Siegfried (2018), Response of Pacific-sector Antarctic ice shelves to the El Niño/Southern Oscillation, *Nature Geoscience*, 11, 121–126, doi:10.1038/s41561-017-0033-0.

Pritchard, H. D., S. R. M. Ligtenberg, H. A. Fricker, D. G. Vaughan, M. R. van den Broeke, and L. Padman (2012), Antarctic ice-sheet loss driven by basal melting of ice-sheets, *Nature*, 484, 502–505, doi:10.1038/nature10968.

Rignot, E., J. Mouginot, M. Morlighem, H. Seroussi, and B. Scheuchl (2014), Widespread, rapid grounding line retreat of Pine Island, Thwaites, Smith, and Kohler glaciers, West Antarctica, from 1992 to 2011, *Geophysical Research Letters*, 421, 3502–3509, doi:10.1002/2014GL060140.

Rignot, E., J. Mouginot, and B. Scheuchl (2011), Ice Flow of the Antarctic Ice Sheet, *Science*, 333, 1427-1430, doi:10.1103/PhysRevB.60.7764.

Rignot, E., S. Jacobs, J. Mouginot, and B. Scheuchl (2013), Ice-shelf melting around Antarctica, *Science*, 341(6143), 266–270, doi:10.1126/science.1235798.

Schaffer, J., R. Timmermann, J.E. Arndt, S. S. Kristensen, C. Mayer, M. Morlighem and D. Steinhage (2016), A global, high-resolution data set of ice sheet topography, cavity geometry, and ocean bathymetry, *Earth System Science Data*, 8, 543-557, doi: 10.5194/essd-8-543-2016.

Schmidtko, S., K. J., Heywood, A. F. Thompson, and S. Aoki (2014), Multidecadal warming of Antarctic waters, Science, 346, 1227–1231, doi:10.1126/science.1256117.

Shepherd, A. et al. (2018), Mass balance of the Antarctic Ice Sheet from 1992 to 2017, *Nature*, 558, 219-222, doi:10.1098/rsta.2006.1792.

Steig, E. J., Q. Ding, D. S. Battisti, and A. Jenkins (2012), Tropical forcing of Circumpolar Deep Water Inflow and outlet glacier thinning in the Amundsen Sea Embayment, West Antarctica, *Annals of Glaciology*, 53, 19–28, doi:10.3189/2012AoG60A110.

Thoma, M., A. Jenkins, D. Holland, and S. Jacobs (2008), Modelling Circumpolar Deep Water intrusions on the Amundsen Sea continental shelf, Antarctica, *Geophysical Research Letters,* 35, L18602, doi:10.1029/2008GL034939.

Turner, J., A. Orr, G. H. Gudmundsson, A. Jenkins, R. G. Bingham, C-D Hillenbrand, and T. J. Bracegirdle (2017), Atmosphere-ocean-ice interactions in the Amundsen Sea Embayment, West Antarctica, *Reviews of Geophysics*, 55, 235-276, doi:10.1002/2016RG000532.

Vaughan, D.G., J. C. Comiso, I. Allison, J. Carrasco, G. Kaser, R. Kwok, P. Mote, T. Murray, F. Paul, J. Ren, E. Rignot, O. Solomina, K. Steffen, and T. Zhang (2013), Observations: Cryosphere, in: *Climate Change 2013: The Physical Science Basis. Contribution of Working Group I to the Fifth Assessment Report of the Intergovernmental Panel on Climate Change*, T. F. Stocker, D. Qin, G.-K Plattner, M. Tignor, S. K. Allen, J. Boschung, A. Nauels, Y. Xia, V. Bex, and P. M. Midgley (eds.), Cambridge University Press, Cambridge, United Kingdom and New York, NY, USA, 65pp.

Walker, C. C., and A.S. Gardner (2017), Rapid drawdown of Antarctica's Wordie Ice Shelf glaciers in response to ENSO/Southern Annular Mode-driven warming in the Southern Ocean, *Earth and Planetary Science Letters*, 476, 100–110, doi:10.1016/j.epsl.2017.08.005.

Webber, B. G. M., K. J., Haywood, D. P. Stevens, P. Dutrieux, E. P. Abrahamsen, A. Jenkins, S. S. Jacobs, H. K. Ha, S. H. Lee, and T. W. Kim (2017) Mechanisms driving variability in the ocean forcing of Pine Island Glacier, *Nature Communications*, 8(14507), doi:10.1038/ncomms14507.

Weertman, J. (1974), Stability of the junction of an ice sheet and an ice shelf, *Journal of Glaciology*, 13(67), 3–11.

Wouters, B., A. Martin-Español, V. Helm, T. Flament, J. M. van Wessem, S. R. M. Ligtenberg, M. R. van den Broeke, and J. L. Bamber (2015), Dynamic thinning of glaciers on the Southern Antarctic Peninsula, *Science*, 348, 899–903, doi:10.1126/science.aaa5727.

Zhang, X., A. F. Thompson, M. M. Flexas, F. Roquet, and H. Bornemann (2016), Circulation and meltwater distribution in the Bellingshausen Sea: From shelf break to coast, *Geophysical Research Letters*, 43, 6402–6409, doi:10.1002/2016GL068998, 2016.

---

## Referee Comment (RC2) · Anonymous Referee #2 · 13 Apr 2019

**1  Summary:**

This paper is a direct response to one of the geoengineering solutions presented in Moore et al 2018. The authors use a global ocean model to determine how an engineered submarine wall in the Amundsen Sea might affect the integrated basal melt rate of Antarctic ice shelves. They find that, as deep warm water is prevented from entering the ice shelf cavities of the Amundsen Sea, basal melt rates in other locations are increased. The net effect of the submarine wall is an integrated 10% decrease in basal melt from a control case.

[Figure]

**2 Recommendation:**

This paper is timely and well-written. It provides a concise response to an earlier publication (Moore et al 2018), and illustrates one specific aspect of the complexity of geoengineering problems. Namely, that if heat is blocked from melting certain ice shelves, it may well go elsewhere and cause additional problems. There are many ways this paper could be expanded, however, the authors have done well in isolating and investigating a specific problem and containing it to a brief comment. I recommend publication after a few minor modifications.

**3 General comments:**

I strongly encourage the authors to make the data used for this paper freely available online. Ideally, this would include model source code, forcing files, and code used for analysis. At the least, the authors should deposit a subset of model output used for the calculations in a repository such as Zenodo.

There are several interesting points of discussion the authors do not address. As this paper is a direct comment on Moore et al 2018, it is up to the authors if they want to include further speculation. Here I list several of these points, for both the authors and the broader community to consider:

- Where does the heat go? If the integrated basal melting loss around Antarctica is 10% lower in the WALL experiment, the heat that would have caused melt must be somewhere else. Is it still in the ocean (and what part), transferred to the atmosphere or to sea ice melt, or somewhere else?

- The model uses forcing for present day conditions. Under future warming scenarios, would the WALL experiment cause more or less integrated basal melt?

- Both this paper and Moore et al 2018 focus on sea level rise and investigate geoengineering possibilities that may also affect aspects of the climate outside of sea level. In this case of a wall, oceanic heat is redistributed and water circulation in ice shelf cavities in the Amundsen Sea is reduced. This has implications for the local ecosystem that is controlled by ice shelf cavity circulation (e.g., St. Laurent et al 2017). It would be useful to note that geoengineered 'solutions' to sea level rise affect more than just the target area or problem, as the climate system is not separable into individual pieces.

**4  Line comments:**

Line 58-60 - It would be useful here to know that the WALL in your simulations does not match the wall in Moore et al 2018. Option to include a brief discussion why, or refer the reader to a later section.

Line 81 - Remove the fraction; use only the 85% value

Line 135+ - A portion of the references are redundant.

**5  References:**

Moore, J. C., Gladstone, R., Zwinger, T., Wolovick, M. (2018). Geoengineer polar glaciers to slow sea-level rise. Nature, 555(7696), 303-305.

St-Laurent, P., P. L. Yager, R. M. Sherrell, S. E. Stammerjohn, and M. S. Dinniman (2017), Pathways and supply of dissolved iron in the Amundsen Sea (Antarctica), J. Geophys. Res. Oceans, 122, 7135–7162, doi:10.1002/2017JC013162.

---

## Author Comment (AC1) · 7 Jun 2019

**Comment on, "Brief Communication: A submarine wall protecting the Amundsen Sea intensifies melting of neighboring ice shelves", by Gürses et al.**

Mike Wolovick

Our reply is written in blue.

First of all, I would like to congratulate the authors on an excellent modeling paper answering a concise scientific question. This sort of skeptical engagement from the scientific community is exactly what we were hoping for when we wrote our original geoengineering papers. It is vitally important that all potential side effects of any geoengineering proposal are explored thoroughly, including side effects that the original authors did not think of. This side effect certainly falls under that category; we did not anticipate that blocking warm water from reaching some ice shelves would cause it to increase melting at other shelves.

Thank you very much for your encouraging comments.

However, I am worried that casual readers might draw the implication that an intervention which merely redirected melting from one ice shelf to another would therefore be ineffective. It is important to emphasize that what matters from the perspective of human societies is not the floating ice shelves, which already displace their weight in water and thus make no direct contribution to sea level when they melt, but rather the grounded ice, which can raise sea levels if it flows into the ocean. The floating shelves are important only insofar as they act to buttress the grounded ice and prevent grounding line retreat. The authors themselves alluded to this issue in the conclusion, writing, "[I]t is an open question if this triggers Marine Ice Sheet Instability in the other shelves, because the stability depends on the distribution of pinning points, sloping of the bed, depth and width of submarine troughs, and the softness of the bed, for instance. The onshore bed properties of the western Marie Byrd Land, where Pine Island and Thwaites Glaciers are located, are probably more favorable for a stable situation than in the eastern Marie Byrd Land sector."[1]

Thanks for indicating that this could be misunderstood. We have clarified this point.
"The onshore bed properties of the western eastern Marie Byrd Land, where Pine Island and Thwaites Glaciers are located, are most likely vulnerable to the Marine Ice Sheet Instability. Numerous modelling studies show a relic ice cap in the western Marie Byrd Land on the elevated bed rock topography even after the part of the West Antarctic Ice Sheet (WAIS) has collapsed (e.g. DeConto and Pollard, 2016; Feldmann and Levermann, 2015; Golledge et al., 2015; Winkelmann et al., 2015). Hence the western Marie Byrd Land is probably more favorable for a stable situation."
* * *
[1] This statement needs to switch western and eastern, and it is somewhat ambiguously worded at the end. It would be more accurate to say that the onshore bed properties in eastern Marie Byrd Land, where Pine Island and Thwaites Glaciers are located, are more favorable for a runaway instability than western Marie Byrd Land.

I would argue that the above statement is far too weak. We can be highly confident that Pine Island and Thwaites Glaciers are more vulnerable to a runaway Marine Ice Sheet Instability than the areas of western Marie Byrd Land onshore of the Getz Ice Shelf. At the simplest level, we can look at the basal topography of these areas of the ice sheet (Fretwell et al., 2013). There is a large overdeepened marine basin onshore of Thwaites Glacier, but there is elevated basal topography, in some places above sea level, onshore of Getz (Fig 1). Thwaites and Pine Island are also retreating at the present day (Turner et al., 2017), and indeed, there have been reasonable suggestions from both data and models that they have already begun a runaway retreat (Favier et al., 2014; Joughin et al., 2014; Rignot et al., 2014). Furthermore, the geographic position of Thwaites and Pine Island ensures that a runaway retreat there will trigger a general collapse of West Antarctica through a "backdoor" destabilization of the Filchner-Ronne and Ross sectors (Feldmann and Levermann, 2015).

By contrast, ice sheet models almost always show that the ice cap onshore of western Getz is the most stable part of WAIS. The elevated basal topography there (Fig 1) allows a relic ice cap to persist even after the rest of WAIS has collapsed. This relic ice cap can be seen in DeConto and Pollard (2016); Winkelmann et al. (2015); Golledge et al. (2015); Feldmann and Levermann (2015); and in Pollard and DeConto (2009). I have compiled snapshots of ice sheet geometry from all of those models in Fig 2. The relic ice cap in western Marie Byrd Land is a robust feature of ice sheet models because all of the models are responding to the elevated basal topography in that region. Based on a convergence of evidence from basic MISI theory, observations, and models, we can have a high degree of confidence that Pine Island and especially Thwaites are the most unstable parts of West Antarctica, while the ice cap in western Marie Byrd Land is the most stable part. In fact, that little ice cap is likely to be the last thing left standing long after the rest of WAIS has collapsed.

We hope that the clarified sentences address your concern.

The model results presented in this paper indicate that a wall built across the Amundsen Sea Embayment at depth could successfully trade high melt rates at Pine Island and Thwaites for high melt rates at Getz. The current state of glaciological knowledge strongly indicates that the ice cap onshore of the western Getz Ice Shelf is the most stable part of WAIS, and the overdeepened topography of Pine Island and Thwaites are the most unstable parts[2]. While it is always important to quantify all side effects of a potential geoengineering project, not all ice shelves are created equal in their importance to ice sheet stability and sea level rise. In my opinion, a geoengineering effort that shifted high melt rates from the most unstable part of WAIS to the most stable part of WAIS would be a smashing success. From the perspective of humanity's interest in a stable sea level, the trade described by this paper is an excellent one.

**References**
* * *
[2] Although it is also important to be open to the possibility that our consensus understanding may be wrong. In particular, I worry that a lack of high density ice thickness measurements in western Marie Byrd Land could be hiding deep subglacial troughs and therefore causing us to overestimate the stability of that region.

DeConto, R. M., & Pollard, D. (2016). Contribution of Antarctica to past and future sea-level rise. *Nature*, *531*(7596), 591–597. https://doi.org/10.1038/nature17145

Favier, L., Durand, G., Cornford, S. L., Gudmundsson, G. H., Gagliardini, O., Gillet-Chaulet, F., et al. (2014). Retreat of Pine Island Glacier controlled by marine ice-sheet instability. *Nature Clim. Change*,*4*(2), 117–121. https://doi.org/10.1038/nclimate2094

Feldmann, J., & Levermann, A. (2015). Collapse of the West Antarctic Ice Sheet after local destabilization of the Amundsen Basin. *Proceedings of the National Academy of Sciences*, *112*(46), 14191. https://doi.org/10.1073/pnas.1512482112

Fretwell, P., Pritchard, H. D., Vaughan, D. G., Bamber, J. L., Barrand, N. E., Bell, R., et al. (2013). Bedmap2: improved ice bed, surface and thickness datasets for Antarctica. *The Cryosphere*, *7*(1), 375 –393. https://doi.org/10.5194/tc-7-375-2013

Golledge, N. R., Kowalewski, D. E., Naish, T. R., Levy, R. H., Fogwill, C. J., & Gasson, E. G. W. (2015). The multi-millennial Antarctic commitment to future sea-level rise. *Nature*, *526*(7573), 421– 425. https://doi.org/10.1038/nature15706

Joughin, I., Smith, B. E., & Medley, B. (2014). Marine Ice Sheet Collapse Potentially Underway for the Thwaites Glacier Basin, West Antarctica. *Science*, *344*(6185), 735–738.https://doi.org/10.1126/science.1249055

Pollard, D., & DeConto, R. M. (2009). Modelling West Antarctic ice sheet growth and collapse through the past five million years. *Nature*, *458*(7236), 329–332. https://doi.org/10.1038/nature07809

Rignot, E., Mouginot, J., Morlighem, M., Seroussi, H., & Scheuchl, B. (2014). Widespread, rapid grounding line retreat of Pine Island, Thwaites, Smith and Kohler glaciers, West Antarctica from 1992 to 2011. *Geophysical Research Letters*, *41*(10), 3502–3509. https://doi.org/10.1002/2014GL060140

Turner, J., Orr, A., Gudmundsson, G. H., Jenkins, A., Bingham, R. G., Hillenbrand, C.-D., & Bracegirdle, T. J. (2017). Atmosphere-ocean-ice interactions in the Amundsen Sea Embayment, West Antarctica. *Reviews of Geophysics*, *55*(1), 235–276. https://doi.org/10.1002/2016RG000532

Winkelmann, R., Levermann, A., Ridgwell, A., & Caldeira, K. (2015). Combustion of available fossil fuel resources sufficient to eliminate the Antarctic Ice Sheet. *Science Advances*, *1*(8), e1500589.https://doi.org/10.1126/sciadv.1500589

---

## Author Comment (AC2) · 21 Jun 2019

**Review of "Brief Communication: A submarine wall protecting the Amundsen Sea intensifies melting of neighboring ice shelves" by Gürses et al., 2019**

Our reply is written in blue.

**Summary**

The authors use an ice-ocean model to investigate the effects of a submarine wall on the basal melting of the ice shelves fringing the Amundsen Sea Sector, West Antarctica. While a clear reduction in basal melting shoreward of (and in some cases adjacent to) the wall is detected, an enhanced melting signal is also found along the neighboring Getz Ice Shelf (as well as farther afield at George VI and Amery Ice Shelves), which the authors state may reduce the effectiveness of such a construction. However, despite increased melting across these regions, the large reduction in melting simulated over the Amundsen Sea Sector is believed to contribute to a ~10% decrease in Antarctica's total mass loss. Raising important questions about the usefulness (or otherwise) of geoengineering as a means to mitigate Antarctic ice-mass loss, I therefore believe the findings presented in this manuscript are timely and will be of genuine interest to the readership of The Cryosphere. However, prior to publication, I would encourage the authors to address several important points detailed below.

Thank you very much for your encouraging comments. We are also happy that your engagement and healthy skepticism helps to improve the manuscript significantly.

**General comments**

*Model bathymetry*

In Section 2, the authors detail the construction of the wall in their model, which acts to block the intrusion of circumpolar deep water (CDW) onto the Amundsen Sea's continental shelf. While I unfamiliar with the technicalities of the FESHOM model, I was very surprised to see the use of RTOPO1 in the model setup for bathymetry, ice shelf geometry and grounding line location. This product has now been superseded by at least 3 updated bathymetric models (e.g. Bedmap2 (Fretwell et al., 2013); IBCSO (Arndt et al., 2013); RTOPO2, Schaffer et al., 2016)), which have significantly improved our understanding of the Amundsen Sea Sector's continental shelf and sub-ice shelf cavity geometry via a range of new in-situ observations and model predictions. A simple subtraction of RTOPO1 from IBCSO (Figure 1 of this review) emphasizes this point, and shows substantial between-model differences in bedrock elevation throughout the domain, including underneath the ice shelves.

During the discussion of our results in the "Conclusion" section, we have added some paragraphs highlighting the limitation of our simulations clearly. Regardless of this important aspect, we a confident that our main findings are robust: a) A wall shielding the Amundsen Sea Embayment reduces basal melting rates within the protected region, b) the rejected warm water masses flows along the wall westward, c) west of the wall warmer water masses drive enhanced basal melting. Please see also our reply after the next paragraph.

It is conceivable that these differences may lead to substantial variations in modelled CDW ingress and basal melting throughout the Amundsen Sea Sector, which may in turn have impacts for the corresponding Antarctic-wide melt budgets presented in Figure 3, and potentially the overall conclusions of the paper. In order for the findings of this paper to be convincing, I therefore strongly encourage the authors to rerun their analyses using one or all of these models, and carefully adjust the figures/text as necessary to incorporate any new or additional results.

I get the impression that we favor different aspects of the performed work and what shall be main message. Unfortunately, we disagree here. As stated above and now discussed in some detail in the extended "Conclusion" section, we are quite confident about our main finding: The wall protects the Amundsen Sea and redirects the warm water westward where we detect enhanced basal ice shelf melting. We agree it might be important to analyze how different bedrock topographies / bathymetries impact our results. But our focus highlights the overlooked side effect of the proposed targeted geoengineering: A wall rejecting the flow of warm water diverts these warm water masses to a different location and amplifies ice loss there. Theoretical dynamical principles (flow follows geostrophic [f/h] contours due to conservation of potential vorticity) support this described findings of our model simulations. We decide to restrict the current study to this finding and have, therefore, intentionally selected the "Brief Communication" format to convey only this main finding. We are confident that this aspect is new to the glaciology communities as the other anonymous reviewer and the openly left discussion contribution of Mike Wolovick highlights.

*Standard of writing/English language*
While I appreciate that English may not be the native language of the authors, I echo the Editor's initial comments that the main text still includes a large amount of verbose and/or non-standard sentence construction, which at times makes the flow of the manuscript difficult to follow and/or comprehend. This is particularly true of the end of Sections 3 and 4, where the authors concluding statements appear to downplay the importance of intensified neighboring melt - the focus of the title and abstract (see specific comments below), and thus what I initially perceived to be the key message of this research. I have attempted to restructure large parts of the main text to the best of my ability, but prior to publication I would again ask the authors to very carefully read through their manuscript with the assistance of a native English speaker/proofreader, to improve the readability of this otherwise interesting piece of research.

Our finally submitted version had been checked and corrected by a North American native speaker. Anyhow, to improve the quality of the manuscript, we followed most of the technical comments listed below.

*Citations*
Whilst the style of referencing in this manuscript is generally satisfactory, I think the main text is somewhat marred by an over-reliance of modelling-based studies, and omits
a lot of other key research on (e.g. observationally constrained) Amundsen Sector ice-ocean-atmosphere interactions and/or glacial change. Such citations should be added to the text to provide a more reasoned/well-rounded discussion. Occasionally, citations are also omitted from

sentences altogether, which should also be addressed. (See my suggested edits in the specific comments below).

For the submitted article we had to fulfill strict limitations, which are part of the "Brief Communication" format. Here I cite the essential sentence: "**Brief communications have a maximum of 3 figures and/or tables, a maximum of 20 references, and an abstract length not exceeding 100 words**." Please note that the bold characters come from the provided text template obtained from the "The Cryosphere" webpage. They probably highlight the importance of these limits. Anyhow, during the review we follow the reviewers partly and exceed the reference limits, but we still try to use less that than the suggest amount of references to write a short article following the idea behind the "Brief Communication" format. We also break the four page limit, after all the suggested additions by the reviewers and comments from the community have been taken into account.

*Introduction*
At the end of the introduction section, I think some words on the flaws and critical 'next steps' of the studies presented by Moore et al. (2018) and Wolovik and Moore (2018) should be added, to qualify the present study and emphasize to the reader why modelling the impacts of building such a wall might be required. The inclusion of a sentence similar to the one on Lines 116-117 could also be added to contextualize the wider role of geoengineering, and hence the need to accurately predict 'adverse side effects'.

We followed your suggestion (even if we exceed the four page limit) and we added:
"In this paper we investigate how a submarine wall, shielding the Amundsen Sea Embayment (Figure 2a), reduces the basal melting of rates ice shelves flowing into the Amundsen Sea Embayment. The warm water masses rejected by the wall enhance ice shelves west of the wall. These effects counteract the wall's purpose mitigating sea level raise. In this study, we neglect feedbacks between changes of basal melting rates and advance or retreat, respectively, of impacted ice shelves. We do not analyze how the wall hinders the exchange of nutrients and influences submarine biological processes."

*Section 3 (Lines 63-64)*
Following Section 2 (Lines 56-57), are your modelled 1947-2007 ocean temperatures also restricted to summertime means? Or do they reflect annual averages? I think this might be worth explicitly stating here. Similarly, if indeed they do reflect annual averages, then have you also considered the importance of seasonal changes in CDW ingress onto the continental shelf, as has been noted in the recent literature? (e.g. Thoma et al., 2008; Steig et al., 2012; Dutrieux et al., 2014; Webber et al., 2017). Such changes may lead to large variations in bottom temperatures over seasonal timescales (and hence basal melt rates), which may not be representative of the in-situ temperatures shown in Figure 1 of the manuscript. If this is the case, then what steps have been taken to validate the temperatures estimated by your model during non-summer seasons?

We are sorry that we have been misunderstood, but we show only potential temperatures to avoid those differences in the ocean depths of individual observations influence the presented

difference (Figure 1) between simulated and observed temperatures. Regarding the indeed correctly highlighted importance of the seasonality, we have modified text to clarify this point: "Considerable oceanic variability has been detected at both seasonal and interannual timescales in front of both Pine Island (Webber et al., 2017) and Dotson Ice Shelf, located between Thwaites Glacier and Getz Ice Shelf, (Jenkins et al., 2018), for instance. It is driven by both local and remote forcing. Hence we shall expect some differences between merged hydrographic observations and a simulated long-term mean, while a reliable climatological data set is lacking for our region of interest. Therefore, we use existing observations for comparison with our simulations under the assumption that available observations represent a quasi-mean state."`

**Specific scientific comments**

Ln 74 – "The warm water mass penetrates through the Getz Ice Shelf into the walled region". Following my concerns on the use of RTOPO1 above, is this phenomenon present when the model is run with more updated cavity geometry information (e.g. IBCSO/RTOPO2)? Equally, what impact does this have on the simulated spatial distribution and magnitude of melting of Abbot Ice Shelf? In Figure 1 of this review, it is apparent that significant (> +/- 250 m) differences exist underneath these ice shelves, so I would encourage the authors to give this careful consideration.

As stated above, we have added several paragraphs discussing the limitations of our study in the "Conclusion" section.

Lns 85 to 87 – These sentences appear highly speculative and in physical terms, I don't understand how this could be the case. The positioning of the ACC over the Bellingshausen Sectors' continental shelf break has been implicated as the predominant driver of unmodified CDW flooding across this region (e.g. Holland et al., 2010; Bingham et al., 2012; Schmidtko et al., 2014; Wouters et al., 2015; Paolo et al., 2015; Christie et al., 2016; Zhang et al., 2016; Hogg et al., 2017), which is presumably the overriding driver of melt variability at GVIIS. As such, I don't understand how mCDW, which would presumably be constantly freshening during its transport underneath and eastward of the Abbot Ice Shelf, could either reach GVIIS or play a more important role than the influence of the ACC here. I would encourage the authors to carefully consider this point and either clarify why they think this to be the case, and/or amend the text/interpretations as necessary.

Thanks for indicating this issue. We discuss in the "Conclusion" section limitations of our simulations and highlight that this features are not robust and may vanish if we would run simulations coupled to an interacting atmosphere.

The same comment applies to why they think reductions in melt rate in the Amundsen Sector may influence melting at Amery Ice Shelf. Presumably any propagation in the coastal current would become entrained within the Ross Gyre, and not extend to the other side of the continent (cf. Nakayama et al., 2014; Dotto et al., 2018)? Assuming it did, however, then presumably any diverted CDW would again be freshened during its advection towards these regions? As above, I'd like to see a more convincing discussion of why the authors believe this to be the case added here.

I am also interested to see how these findings may change when the model is forced with more updated bathymetry as discussed above. While Figure 1 in this review only shows the Amundsen Sea Sector and its surrounds, significant differences in bathymetry also exist around the continent.

We discuss it the extended "Conclusion" section. Please see also the former reply above.

**Technical comments**

Title – For those unfamiliar with the geography of Antarctica, I would reword the title to "A submarine wall protecting the Amundsen Sea, West Antarctica, intensifies melting of neighboring ice shelves" or similar.

We think the title is appropriate. We followed your suggestion added in the very first sentence of this work "West Antarctica." See next reply please.

Ln 8 – Add "Sector of West Antarctica" after 'Amundsen Sea'. Also reword the end of the sentence to "…acceleration of ice discharge from upstream grounded ice" for technical accuracy.

Our original abstract should only contain 300 words. However we like to follow your suggestion and improve the quality.

Ln 9 – 'et al' is a Latin abbreviation for 'et alia', and so a period should follow the 'al' (i.e. 'et al.'). I have noticed this small error throughout the manuscript, so the authors should address this universally throughout the document. Also, add the word 'ocean' between 'warm water'.

Thanks for indicating it. We have relied blindly on a commercial product to organize our literature. We have manually checked and adjust these citations.

Ln 10 – Suggest rephrasing the end of this sentence to "…into the sub-surface cavities of these ice shelves could reduce this risk". The word 'sea' preceding 'ice-ocean' model is not needed, and should be removed.

We rephrase as suggested. But we disagree about "sea ice". Since we use a coupled sea ice-ocean model that resolves ice shelves and includes the ice shelf-ocean interaction, replacing "sea ice" by "ice" may raise the question, if we have missed this important climate component.

Ln 11 – Change 'warm water' to 'this water'. Rephrase next sentence to begin "However, these water masses get redirected … which reduces the net effectiveness …".

Rephrased:
"However, these warm water masses get redirected … which reduces the net effectiveness ..."

Ln 14 – Should read "… the warming of Earth's climate is sea level rise". Add a reference to the IPCC (e.g. Vaughan et al., 2013) to the end of the next sentence.

We follow your suggestion.

Ln 15 – Suggest rewording to "Currently, the main ... mean sea levels are the thermal expansion of the world's oceans, the mass losses emanating from the Greenland Ice Sheet, and the world-wide recession of mountain glaciers and ice caps...".

Done.

Ln 17 – Suggest rewording to "... and the ice mass losses originating from the Antarctic Ice Sheet... although Antarctica's...". (Note here the capitalization of the pronoun 'Antarctic Ice Sheet'). At the end of this sentence, a reference to Shepherd et al. (2018) should also be added.

During writing our manuscript we had a hard time fulfilling the limit of 20 references. In our very first manuscript version we had more than half-dozen references short paragraph describing the current sea level contributions. I'm happy to add some of them (such as the Shepherd et al. (2018)) reference.

Ln 20 – Suggest rewording this sentence to read "In Antarctica, remotely sensed, modelled and palaeoclimatological-proxy data indicate that the highest potential for sea level rise will come from the West Antarctic Ice Sheet (Joughin and Alley, 2011), particularly from the Amundsen Sea Sector, where the progressive thinning of its ice shelves over the past ~25 years has greatly enhanced rates of ice mass loss emanating from this sector" or similar. At the end of this sentence, cite e.g. Pritchard et al. (2012); Mouginot et al. (2014); Rignot at al. (2014); Paolo et al., 2015; Shepherd et al. (2018).

Done

Ln 22 – Suggest rewording next sentence to something like: "Here, warm, high salinity circumpolar deep water (hereafter CDW) has been observed to flow onto the continental shelf and flood the cavities underneath the Amundsen Sea Sector's ice shelves, driving high rates of basal melting". Add citations (e.g. Jenkins et al., 2010; Pritchard et al., 2012; Rignot et al., 2013; Jacobs et al., 2013; Depoorter et al., 2013) here.

Done

Lns 25-26 – Merge these two sentences for brevity. Could read something similar to: "Various processes... ice shelf cavities, including, most predominantly, wind-driven changes in Ekman transport, whereby variations in offshore wind stresses lift CDW onto the continental shelf". An abundance of new literature has been published on this phenomenon in recent years, which could/should be cited here in addition to work by Kim et al (2017). These include, but are not limited to: Thoma et al. (2008); Steig et al. (2012); Jacobs et al. (2013); Dutrieux et al. (2014); Walker et al. (2017); Christie et al. (2018); Greene et al. (2018) and Paolo et al. (2018).

As already indicated above, we have a limit of only 20 references. I'm happy to go beyond this strict limit, but we would like to follow the idea behind this limit (having short and concise article) and try to keep a short reference list.

Ln 27 – Suggest rewrite to: "During its transport onto the continental shelf, this water mass is … by mixing with local, fresher on-shelf water masses". A citation is also needed here (suggest Webber et al. (2017)).

Done.

Lns 25-29 – Somewhere in this section I think a short sentence should be added detailing the important role submarine troughs play in amplifying the transmission of CDW to the grounding line (following e.g. Nitsche et al. (2007); Bingham et al. (2012); Dutrieux et al. (2014)). The addition of this sentence would critically also give context to the discussion presented in Section 3 (Line 62).

Modified a former sentence, so that we read now:
"Various processes control the flow of warm water masses (a body of ocean water with a common formation history and a defined range of tracers, such as temperature and salinity, is called water mass) predominately via glacially scoured submarine troughs (Bingham et al., 2012; Dutrieux et al., 2014) into the ice shelf cavities"

Ln 26 – Suggest reworking the rest of this paragraph to the following or similar for conciseness: "In the Amundsen Sea Sector, decadal-scale changes in the draft and intensity of CDW incursion onto the continental shelf – and ultimately the basal melting of the ice masses fringing this sector of Antarctica - have also been directly linked to changes in global-scale atmospheric circulation, including the influence of ENSO-induced atmospheric wave trains propagating towards this region from the central tropical Pacific Ocean (Steig et al., 2012; Dutrieux et al., 2014; Jenkins et al., 2018; Nakayama et al., 2018; Paolo et al., 2018)".

Done.

Ln 32 – Suggest the amalgamation of this and the following sentence for conciseness. Could read something like: "Since the West Antarctic Ice Sheet resides on retrograde sloping topography (Mercer, 1978), it is inherently susceptible to a Marine Ice Sheet Instability, whereby the reduced buttressing effect of thinning ice shelves triggers the retreat of upstream ice, leading to larger ice thicknesses at the grounding line (Hughes, 1973; Weertman, 1974; Schoof, 2007)". [Note also here the addition of several classic papers I was surprised to not see in the text. Also, as the term 'grounding line' hasn't been introduced, I would consider also defining this in a short, follow-up sentence].

We followed your suggestion, but we have not added all suggested references, because we shall have a short reference list – as already said, we had originally a very strict limit of 20 references.

Ln 35 – Hyphen required between 'grounding line'. For clarity, next sentence could also be amended to read: "This sustained retreat accelerates the transport of inland ice towards the ocean past the grounding line, where it directly contributes to sea level rise".

Done.

Ln 38 – Full stop required after the abbreviation 'al' as discussed above. Also, suggest changing 'this ice sheet collapse mechanism' to 'marine ice sheet instability' since this has just been defined above.

We followed your text suggestion.

Ln 39 – Suggest changing 'warm water with' to 'CDW via the erection of'.

We wrote "warm Circumpolar Deep Water via the erection of"

Ln 40 – 'Thwaites Glacier' is a pronoun, hence the word 'the' directly preceding it should be omitted. Also suggest reword of the end of this sentence to "…Thwaites Glacier – one of the largest contributors of ice discharge into the Amundsen Sea (Rignot et al., 2011; Mouginot et al., 2014; Turner et al., 2017; Shepherd et al., 2018)" for clarity. [Note the addition of several key recent citations here].

We followed your suggestion, but we have not added all suggested references, because we shall have a short reference list. We restricted the list to the two newest references.

Ln 41 – This sentence is highly repetitive of the preceding sentence explaining the work of Moore et al. (2018), but can easily be fixed by changing to something like: "In addition to the erection of subsurface walls (cf. Moore et al., 2018), they imposed artificial pinning points to enhance the buttressing effect of ice shelves on grounded ice. Both measures were found to successfully reduce ice mass losses emanating from this sector of Antarctica".

Done.

Ln 42 – As noted in my general comments, some words on what these studies didn't examine/consider (i.e. the potentially adverse effects elsewhere), in order to qualify the research presented in this paper, should be added here.

We added text as described above. Please see reply to raised related general comment.

Ln 45 – Should read "Amundsen Sea Sector's ice shelves". Next sentence should also read "…horizontal resolution (minimum 5km) around Antarctica and its … and has 100 vertical levels (z-coordinate)." for clarity.

Done.

Ln 49 – Should references be listed in chronological order? Also suggest rewording following sentence to "While coarse resolution ocean models have been found to underestimate the ocean-induced melting of Antarctica's ice shelves, our basal melting rates are in reasonable agreement

with recent observational estimates". [The authors should also add appropriate citations to the observational estimates they refer to, as well as a cross reference to their Figure 2b here].

We use the suggested rephrasing and we added the reference of the used reference basal melting rates. We have ordered them in alphabetic order as determined by "The Cryosphere" plugin of our reference system.

Ln 52 – Suggest using the word 'of' in place of 'from' for grammatical accuracy. See also my comments above regarding my concerns over the use of RTOPO1.

Done.

Ln 54 – Suggest change to "This forcing period is run twice".

Done.

Lns 58-60 – I think these two sentences could be reworked to become much easier to read/comprehend. Suggest reword to: "We investigate differences in ice shelf basal melting with (WALL) and without (CTRL) the erection of a wall surrounding the Amundsen Sea (Figure 2a)" [see also my comments on the manuscript's figures below]. Then: "This feature follows the approximate location of the continental shelf break (~1000 m), and blocks CDW inflow from the deep ocean onto the Amundsen Sea Sector's continental shelf".

We used instead a slightly modified sentence:
"We investigate differences in ice shelf basal melting with (WALL) and without (CTRL) the erection of a wall surrounding the Amundsen Sea (Figure 2a). This feature follows the approximate location of the continental shelf break, and blocks any circulation below 350 m depth, such as the CDW inflow from the deep ocean onto the Amundsen Sea Sector's continental shelf."

Ln 62 – Suggest amalgamating the first two sentences for clarity and conciseness. "Consistent with oceanographic observations [Authors should add reference to the appropriate citations and/or manuscript figure here], our CTRL experiment simulates accurately the ingress and delivery of mCDW through submarine troughs towards the ice shelves fringing the Amundsen Sea Sector". [Note also that the place name 'Amundsen Sea Embayment' is used here for the first time. This has not been introduced prior to this line, so I would suggest using either 'Amundsen Sea Sector' or 'Amundsen Sea Embayment' universally throughout the manuscript for consistency].

We used the suggested sentence and added a reference to our first figure. In our understanding is the Amundsen Sea Embayment the part of the Amundsen Sea between the wall and the coast. We define this term in the section above (see comment to your suggestion of the former line 42). In Amundsen Sea Sector includes the Amundsen Sea Embayment and the ambient continental shelf region.

Ln 63 – Suggest 'acquired' in pace of 'taken'. I would also consider rephrasing this sentence for clarity to "… acquired in austral summer (cf. Section 2), also strongly agree with the spatial

distribution of our simulated temperatures, giving confidence in our abilities to accurately predict basal melting in the present study" or similar.

We followed your advice.

Ln 65 – This sentence is highly verbose, and could be shortened considerably. Suggest something like: "Contrary to our CTRL experiment, our erected wall blocks the ocean below 350 m depth and suppresses the direct inflow of CDW to the interior of the Amundsen Sea".

Done.

Ln 67 – Change '(Figure 2)' to '(Figure 2 a)' for clarity of reading/reference to figures [see also my comments on the manuscript's figures below]. I also suggest restructuring the following sentence to "Enhanced sea ice formation is also simulated, enabled by a resulting colder water column and the consequent release of brine into the underlying ocean across this region".
Ln 68 – I found the context of this sentence almost impossible to comprehend without reading the next paragraph, so I'd suggest rewording to the following, and also inserting a cross reference to Figure 2. Sentence could read something like: "However, despite the brine-induced salinification of the water column here, this phenomenon is insufficient to maintain the pronounced melt rates observed in the presence of unobstructed mCDW inflow (cf. Figure 2),
as discussed below". [NB.: brine is by definition salty, hence the inclusion of the word 'salty' is superfluous].

We write:
"This colder water column supports enhanced sea ice formation, which releases brine into the underlying ocean across this region. However, the brine-induced salinification is insufficient to compensate the salinity supply of the unobstructed mCDW inflow."

Ln 70 – The construction of this sentence is again rather difficult to comprehend, and can be simplified by saying something like: "…, which lies shoreward of the easterly Antarctic Coastal Current residing over the continental shelf break at this location". [Note: A citation should also be added here].

We deleted the subordinate clause.

Ln 71 – Suggest changing the word 'through' with 'via'.

Done.

Ln 72 – Suggesting rephrasing part of this sentence to "the Abbot Ice Shelf's sub-ice shelf cavity (south of Thurston Island) contributes to this cooling (Figures 2a and b)". [Note also the added cross reference to Figures 2a and b].

Done.

Ln 72 (sentence beginning "The deflected …") – Suggest changing the beginning of this sentence to "Seaward of this wall, mCDW …", and amalgamating this and the next sentence together. (At present, they are highly repetitive, and could easily be reformulated into one concise statement).

Ln 76 – Add reference to your Figures 2b and c. In the next sentence, add a comma after 'However'.

Done.

Ln 77 – For ease of reading/cross reference to your Figure 2, I would suggest changing the contents of the parentheses to "(central and western Getz Ice Shelf; Figure 2c)".

Done.

Ln 78 – Add a comma after the word 'therefore', remove the comma after 'mass', and add the word 'have' prior to 'impacted'. Also suggest changing the word 'fringing' to 'neighboring' in line with the manuscript's title.

Thanks and Done.

Ln 80 – "longitudinal dependence". I'm not sure this is the correct term, given that longitude itself does not directly contribute to the basal melting of ice. 'Longitudinal distribution' would perhaps be more suitable. Also, at the end of this sentence, I suggest the authors add "… Antarctica, with and without the erection of the submarine wall" for clarity.

Done.

Ln 81 – Embayment or Sector? See my comment re: Ln 62. Also suggest merging the end of this and the next sentence to: "In the Amundsen Sea Sector [Embayment?], ice mass losses around Pine Island Glacier drop by 85%. This phenomenon contrasts with the increased ice mass loss observed at Getz Ice Shelf as discussed above (see also Figure 2c), where melting increased by ~50%.".

We follow your advice but we use "ice mass loss detected at Getz Ice Shelf" to avoid that any reader misunderstands "observed".

Ln 83 (sentence beginning "In the western Bellingshausen Sea") – This sentence is highly repetitive of the content discussed in Lines 70-74, so could easily be removed or integrated with Lines 70-74.

We shorted it drastically: "As discussed above, basal melting is reduced in the western Bellingshausen Sea."

Ln 85 – Suggest rewording this sentence to "In addition to the decreased melting simulated underneath Abbot Ice Shelf, basal melting at George VI Ice Shelf increased by up to 10%.". [Note

also that the GVIIS resides on the western flank of the Antarctic Peninsula, not west of the Peninsula].

We followed your suggestion.

Ln 87 – Add a comma after 'East Antarctic Ice Sheet'.

Done.

Ln 90 – Following my general comment above, the concluding remarks of this sentence are hard to comprehend, and appear to underplay the key message of the title and abstract. Do you mean to say that while localized melting is enhanced across some neighboring ice shelves, these signals are minimal compared with the simulated continent-wide reductions in melt elsewhere? If this the answer to my question is yes, which I suspect to be the case, then
I'd recommend amending the title, abstract and conclusions to provide a more focused argument in favor of this point. In any case, some rephrasing of this sentence is needed to make your conclusions explicitly clear.

We transformed the message into a single paragraph:
"Beside regional changes of the basal melting rates, we inspect the continental-wide integrated effect. The reduced ice loss in the Amundsen Sea Embayment is larger than the corresponding enhanced melting at the western end of the wall. The total ice loss by ice shelves around Antarctica is 10% lower for the WALL experiment."

Ln 94 – Suggest beginning with "In this study, a submarine wall erected along the continental shelf of the Amundsen Sea is found to suppress the inflow of circumpolar deep water onto the continental shelf. This freshens water masses residing shoreward of the wall, resulting in significantly reduced basal melting rates of the ice-shelves located there. However, inflowing CDW seaward of this wall is found to be redirected westward towards Getz Ice Shelf, where it enhances basal melting by up to 50%...".

We follow your suggestion.

Lns 98-101: Like the concluding remarks of Section 3, it is difficult to understand with absolute certainty what the key take home message is from these sentences. Is it the fact that the melting enhances in neighboring regions as a result of constructing a wall, or that these enhanced melting signals are minimal when compared to the Antarctica's overall mass budget? The authors should rephrase this section to make this explicitly clear. Also, given the opening sentences of the conclusion, there is a lot of redundancy/repetition on how CDW is diverted to Getz and causes enhanced losses in this section, which should be removed.

We rephrase it:
"Hence the wall reduces the ice loss of the most vulnerable ice shelves along the margin of the Western Antarctic Ice Sheet, which is not compensated by enhanced melting in the west. Integrated over Antarctica the ice loss decreases by 10 %."

Lns 101-105 – This section comprises mainly of MISI theory, which was covered in the introduction, and so is not required here. I'd recommend removing this entire section, and instead give brief mention to MISI in the following section (see comment below). On a side note, while I suggest this part of the discussion be excised from the text, I also completely disagree that Thwaites and Pine Island Glaciers have the potential to be more stable than the Marie Byrd Land Sector, owing to the deeply bedded, retrograde bed slopes and subglacial basins they reside on (e.g. Bedmap2, RTOPO1, RTOPO2, IBCSO, ALMAP etc.). Also, I presume this sentence contains a typo in that 'eastern Marie Byrd Land Sector' should actually read 'western Marie Byrd Land Sector' (i.e. the region flowing into Getz Ice Shelf)?

Unfortunately, we have indeed mixed up east and west. This part has been changed according to a detailed comment by Mike Wolovick.

Lns 106-115 – The construction of this paragraph is very hard to follow and should be edited to offer a more fluid and concise discussion. I suggest the following rewrite, in this particular order:
1. A very brief summary of what building a wall means in terms of basal melting in the Amundsen Sea Sector (including Getz);
2.  How the findings of this research compare to the ideas presented by Moore et al. (2018), and what the implications of building the shorter wall he discusses would likely be on this region, and then;
3. What the implications of both walls would therefore be in terms of MISI, and Antarctica's future contributions to sea level rise.

We have rewritten the entire paragraph:
"Our results suggest that a too small wall blocking only the water flow in the troughs leading to Pine Island, for instance, might be bypassed by warm water masses. For dynamical reasons the (geostrophic) flow of water masses turns to the left (on the Southern hemisphere), if it is not hindered by a topographic obstacle. Therefore warm water masses might even recirculate into the ostensibly protected area if the wall is too small, as the inflow of warm water masses through the Getz Ice Shelf into the walled region suggests. However if a small wall protects only Pine Island successfully, it may redirect the warm water to neighboring ice shelves with a retrograde bed (for example Thwaites Glacier). There it increases basal melting and may trigger Marine Ice Sheet Instability. The detected poleward shift of westerly winds in the Southern Ocean under global warming (Miller et al., 2006) may shifts also the coast easterly winds along Antarctica's coast poleward, which lifts further the interface of warm water masses (isothermal) along the continental slope (Spence et al., 2014). Ultimately warm water masses could enter the continental shelf directly beside the contemporary path following topographic troughs. Under these circumstances the bypassing of a short wall seems to be inevitable, if the wall does not block the entire Amundsen Sea Embayment."

Ln 116 – In light this paper's findings, I recommend editing the end of this sentence to read "…, but the results of this study suggest that such proposals could have adverse side effects". Then begin the next sentence with something like: "To evaluate the effects of using submarine walls to protect Antarctica's ice shelves in greater detail, the use of fully coupled ice-sheet-shelf-ocean

models should be utilized in future analyses. These models should be of sufficiently high resolution to simulate accurately changes in sub-ice shelf cavity geometry (including grounding-line migration and ice-shelf thinning), as well as the influx of mCDW to these locations".

Thanks for your contribution to improve this manuscript. We followed your suggestion.

Ln 121 – Suggest removing this sentence, as all it serves to do is cast doubt on the validity of the findings presented in this paper!

Done.

Ln 126 – Should read "... for his comments, which greatly improved this manuscript".

Done.

Ln 129 – Should read "contributed to the interpretation of the results and proofreading of the manuscript".

Done.

Ln 137 – The full stop after 'Germany' is not needed here.

Fixed and online source added.

Ln 186 – 'Cryopsh.' Should be changed to 'Cryosphere'.

Changed to "The Cryosphere."

Lns 187-235 – Remove.

We prefer to keep these citations, because we have cited these papers.

Figure 1: comments on Figure – I would suggest rescaling this image (particularly all lon/lat labels and color bar size) to more closely align with the scaling of Figures 2 and 3, as its current scaling looks rather odd in comparison. To assist the reader, it would also be highly beneficial to add the ice shelf limits as thin lines onto this plot, similar to those presented in Figure 2. Being picky, I also dislike the sizing and positioning of the glacier and ice shelf labels, which could easily be resized/positioned to be more aesthetically pleasing. If possible, I'd also suggest rotating the figure 90 degrees to align with the orientation of the polar stereographic plots shown in Figures 2 and 3.

We have rotated the Figure 1, so that all plots of the Amundsen Sea Embayment have the same orientation. For Amundsen Sea Embayment, we a polar stereographic projection, where the main coast line is aligned with the page. This optimizes in our understanding the ratio between covered page space and shown information. We are sorry that you dislike our figures, but we would like to use these optimized figures.

Figure 1: comments on caption – For overall clarity and conciseness, I would suggest rewriting parts of the caption as follows: "Figure 1 – Modelled and observed seafloor ocean potential temperatures in the Amundsen Sea Sector of West Antarctica. Inset shows study location. The plot shows … acquired in 1994 and 2010, respectively".

We follow your suggestion.

Figure 2: comments on figure –
- Each sub-plot should be labelled (e.g. a, b, c) to assist the readability of the text. These changes should then be incorporated into the main text and figure caption as necessary.

  We followed your suggestion and added labels for each subplot.

- I would also add ice shelf outlines to the left panel as their current omission looks odd.

  We have added to the figures 1 and 2 the ice shelf edges as lines.

- I would like to see ice shelf limits also added to the inset map for wider geographical context.

  The inset map contains the coast line, which follows the ice shelf edges. We do not draw the grounding line positions, because the plot would look crowded in our area of interest. This inset map show just help to find the location in respect to Antarctica.

- Why is the wall shown in some plots but not others? Suggest adding it to all plots. For consistency, I also suggest using the same color of dashed line in all plot.

  We only show the wall in plots, where the wall has an impact on the results: temperature anomaly, basal melting anomalies.

- Why does the spatial extent of the wall change between figures? Please show the exact location of the wall as defined in your model in all plots.

  The wall location is identical between the plots and goes from Thurston Island to Siple Island. However we use different line types between the plots (depending on the plots size) to not cover important features while the wall is still clearly visible.

- While the arrangement of the figure is generally satisfactory as is, could the right-hand panels be made bigger (at the slight expense of the left-hand panel's size) by arranging all figures side-by-side in a 1 row x 3 columns fashion? At present, it is quite difficult to see the interesting spatial details contained in the melt maps, which may be remedied by making these figures larger.

  We have produced totally new plot and have taken in account your suggestions.

- Relatedly, I find the ice front positions in the right-hand panels almost impossible to see against the blue color scale, which would be improved by enlarging the plots. Also, I'd suggest making them thicker and/or a different color (e.g. black) to make them easier to visualize.

  Our new figure takes your concerns into account.

- The label for Abbot IS goes off the plot and looks ugly. Suggest writing over 2 lines to neaten this up.

  What is ugly? Sorry, I would like to avoid talking about personal views.

Figure 2: comments on caption – Unlike Figures 1 and 3, the caption of this plot is missing a short opening summary of what the figure shows, which should be added for consistency.

The caption is changed:
"Simulated potential ocean temperature anomaly (WALL – CTRL) Figure2a) and simulated basal ice shelf melting rates in b) and its anomaly c). The subplot 2a) shows the simulated potential ocean temperature anomaly (WALL – CTRL) on the seafloor of the Amundsen Sea Embayment and its adjacent ice shelf cavities. The location of the wall is marked as a dashed line and the embayment region is defined in the map d). The middle subplot b) show the simulated melting rates for the control run (CTRL) and the right subplot c) shows basal melting anomaly (WALL - CTRL). The ice shelf edges are highlighted by solid green lines. The following abbreviations are used: Abbot IS (Abbot Ice Shelf), Pine IG (Pine Island Glacier), Thwaites G (Thwaites Glacier) and Getz IS (Getz Ice Shelf)."

Ln 246 – Using my labelling convection, I'd suggest editing this sentence to read "Figure 2a shows simulated ocean potential temperature anomalies (WALL-CTRL) on the seafloor of the Amundsen Sea and its adjacent ice shelf cavities. The location of the wall is denoted by a dashed line….".

We have added sublabel for subplots as suggested.

Ln 250 – A colon should follow the word 'used' (i.e. "The following abbreviations are used: …").

Thanks for indicating it. Done.

Ln 252 – Suggest shortening the last sentence to "Inset shows
study location and other regions referred to the text". Change all instances of e.g. 'left subplot shows' to new, explicitly labelled equivalents here and in the main text.

Figure 3: comments on figure –
- Why is color scale inverted in this plot relative to Figure 2? This is extremely confusing for the reader, and should be amended. To add to this confusion, the labels associated with the color bar appear to be incorrect, whereby, according to the current caption, red should actually denote "shrink".

We use now the same sign convention for the basal melting anomalies in both figures 2 and 3.

- Suggest changing 'shrink' and 'gain' to 'decreased' and 'increased' melt, respectively.

  We have replaced 'shrink' and 'gain'. We now use 'increase' and 'reduction'.

- Like the right-hand plots in Figure 2, ice shelf outlines should be added to this figure.

  We do not provide this ice shelf margins as an additional line, since they would partly cover the low signal seen in some ice shelves. For orientation we added only for the Filchner-Ronne-Ice Shelf, Ross Ice Shelf and Amery Ice Shelf the shelf ice edges.

- It's very hard to see the spatial detail of melting around Antarctica in the current figure, which is a shame, so I'd also strongly suggest increasing the scale of the center map if possible, or including the addition of inset subplots zoomed over key areas (e.g. GVIIS and Amery Ice Shelf) if not.

  Since we discuss in the final "Conclusion" section that some of the remote melt anomalies may disappear in fully coupled atmosphere-ocean-sea ice-ice shelf simulations, we do to provide these zoomed plots. However, we will certainly keep it in mind for any following study.

- Similarly, given the subtle changes in melting simulated underneath Amery Ice Shelf, it would be helpful to provide a zoom-in inset of the CRTL vs. WALL signals shown in the figure for this region.

Figure 2: comments on caption –
Ln 255 – 'Outer ring' is confusing, so I'd suggest rewording to: "Longitude-specific changes in modelled basal melting with (WALL) and without (CTRL) the presence of the submarine wall are shown as dashed red and solid blue lines surrounding the center map, respectively".

We followed your suggestion.

Ln 257 – "in the center map" is superfluous, and should be removed (it is obvious where the black dashed line is).

Done.

[Your] Figure 1 – Difference between IBCSO and RTOPO1 seafloor bathymetry (red, IBCSO is deeper; blue, shallower). How do these differences (and/or those of e.g. RTOPO2) affect your modelled changes in CDW incursion/basal melting within a) the Amundsen Sea Sector and b) the rest of Antarctica following the erection of the wall?

**References**

Arndt, J. E., H. W., Schenke, M. Jakobsson, F. O. Nitsche, G. Buys, B. Goleby, M. Rebesco, F. Bohoyo, J. Hong, J. Black, R. Greku, G. Udintsev, F. Barrios, W. Reynoso-Peralta, M. Taisei,

and R. Wigley (2013), The International Bathymetric Chart of the Southern Ocean (IBCSO) Version 1.0—A new bathymetric compilation covering circum-Antarctic waters, Geophysical Research Letters, 40, 3111-3117, doi:10.1002/grl.50413.

Bingham, R. G., F. Ferraccioli, E. C. King, R. D. Larter, H. D. Pritchard, A. M. Smith, and D. G. Vaughan (2012), Inland thinning of West Antarctic ice-sheet steered along subglacial rifts, Nature, 487, 468–471, doi:10.1038/nature11292.

Christie, F. D. W., R. G. Bingham, N. Gourmelen, E. J. Steig, R. R. Bisset, H. D. Pritchard, K. Snow, and S. F. B. Tett (2018a), Glacier change along West Antarctica's Marie Byrd Land Sector and links to inter-decadal atmosphere–ocean variability, The Cryosphere, 12, 2461-2479, doi:10.5194/tc-12-2461-2018.

Christie, F. D. W., R. G. Bingham, N. Gourmelen, S. F. B. Tett, and A. Muto (2016), Four-decade record of pervasive grounding line retreat along the Bellingshausen margin of West Antarctica, Geophysical Research Letters, 43, 5741–5749, doi:10.1002/2016GL068972, 2016.

Depoorter, M. A., J. L., Bamber, J. A. Griggs, J. T. M. Lenaerts, S. R. M. Ligtenberg, M. R., van den Broeke, and G. Moholdt (2013), Calving fluxes and basal melt rates of Antarctic ice shelves, Nature, 502, 89-92, doi:10.1038/nature12567.

Dotto, T. S., A. N. Garabato, S. Bacon, M. Tsamados, P. R. Holland, J. Hooley, E. Frajka-Williams, A. Ridout, M. P. and Meredith (2018), Variability of the Ross Gyre, Southern Ocean: drivers and responses revealed by satellite altimetry, Geophysical Research Letters, 45(12), 6195-6204, doi:10.1029/2018GL078607.

Dutrieux, P., J. De Rydt, A. Jenkins, P. R. Holland, H. K. Ha, S. H. Lee, E. J. Steig, Q. Ding, E. P. Abrahamsen, and M. Schröder (2014), Strong Sensitivity of Pine Island Ice-Shelf Melting to Climatic Variability, Science, 343, 174-178, doi:10.1126/science.1244341.

Fretwell, P., et al. (2013), Bedmap2: Improved ice bed, surface and thickness datasets for Antarctica, The Cryosphere, 7, 375–393, doi:10.5194/tc-7-375-2013.

Greene, C. A., D. D. Blankenship, D. E. Gwyther, A. Silvano, and E. van Wijk (2017), Wind causes Totten Ice Shelf melt and acceleration, Science Advances, 3(11), e1701681, doi:10.1126/sciadv.1701681.

Hogg, A. E., A. Shepherd, S. L. Cornford, K. H. Briggs, N. Gourmelen, J. A. Graham, I. Joughin, J. Mouginot, T. Nagler, A. J. Payne, E. Rignot, and J. Wuite (2017), Increased ice flow in Western Palmer Land linked to ocean melting, Geophysical Research Letters, 44(9), 4159–4167, doi:10.1002/2016GL072110.

Hughes, T. (1973), Is the West Antarctic Ice Sheet Disintegrating?, Journal of Geophysical Research, 78(33), 7884-7910, doi:10.1029/JC078i033p07884.

Holland, P. R., A. Jenkins, and D. M. Holland (2010), Ice and ocean processes in the Bellingshausen Sea, Antarctica, Journal of Geophysical Research, 115, C05020, doi:10.1029/2008JC005219.

Holt, T. O., H. A. Fricker, N. F. Glasser, O. King, A. Luckman, L. Padman, D. J. Quincey, M. R. and Siegfried (2014), The structural and dynamic responses of Stange Ice Shelf to recent environmental change, Antarctic Science, 26, 646–660, doi:10.1017/S095410201400039X.

Jacobs, S., C. Giulivi, P. Dutrieux, E. Rignot, F. Nitsche, and J. Mouginot (2013), Getz Ice Shelf melting response to changes in ocean forcing, Journal of Geophysical Research: Oceans, 118, 1–17, doi:10.1002/jgrc.20298.

Jenkins, A., P. Dutrieux, S. S. Jacobs, S. D. McPhail, J. R. Perrett, A. T. Webb, and D. White (2010), Observations beneath Pine Island Glacier in West Antarctica and implications for its retreat, Nature Geoscience, 3(7), 468–472, doi:10.1038/ngeo890.

Jenkins, A., D. Shoosmith, P. Dutrieux, S. Jacobs, T. W. Kim, S. H. Lee, H. K. Ha, and S. Stammerjohn (2018), West Antarctic Ice Sheet retreat in the Amundsen Sea driven by decadal oceanic variability, Nature Geoscience, 11, 733-738, doi:10.1038/s41561-018-0207-4.

Mercer, J. (1978), West Antarctic ice sheet and CO2 greenhouse: a threat of disaster, Nature, 271, 321-325.

Mouginot, J., E. Rignot, and B. Scheuchl (2014), Sustained increase in ice discharge from the Amundsen Sea Embayment, West Antarctica, from 1973 to 2013, Geophysical Research Letters, 41, 1576-1584, doi:10.1002/2013GL059069.

Nakayama, Y., R. Timmermann, C. B. Rodehacke, M. Schroder, and H. H. Hellmer (2014), Modeling the spreading of glacial meltwater from the Amundsen and Bellingshausen Seas, Geophysical Research Letters, 41, 7942–7949, doi:10.1002/2014GL061600.

Nitsche, F. O., S. S. Jacobs, R. D. Larter, and K. Gohl (2007), Bathymetry of the Amundsen Sea continental shelf: Implications for geology, oceanography, and glaciology, G3 Geochemistry, Geophysics, Geosystems, 8(10), doi:10.1029/2007GC001694.

Paolo, F. S., H. A. Fricker, and L. Padman (2015), Volume loss from Antarctic ice shelves is accelerating, Science, 348, 327–331, doi:10.1126/science.aaa0940.

Paolo, F. S., L. Padman, H. A. Fricker, S. Adusumilli, S. Howard, and M. R. Siegfried (2018), Response of Pacific-sector Antarctic ice shelves to the El Niño/Southern Oscillation, Nature Geoscience, 11, 121–126, doi:10.1038/s41561-017-0033-0.

Pritchard, H. D., S. R. M. Ligtenberg, H. A. Fricker, D. G. Vaughan, M. R. van den Broeke, and L. Padman (2012), Antarctic ice-sheet loss driven by basal melting of ice-sheets, Nature, 484, 502–505, doi:10.1038/nature10968.

Rignot, E., J. Mouginot, M. Morlighem, H. Seroussi, and B. Scheuchl (2014), Widespread, rapid grounding line retreat of Pine Island, Thwaites, Smith, and Kohler glaciers, West Antarctica, from 1992 to 2011, Geophysical Research Letters, 421, 3502–3509, doi:10.1002/2014GL060140.

Rignot, E., J. Mouginot, and B. Scheuchl (2011), Ice Flow of the Antarctic Ice Sheet, Science, 333, 1427-1430, doi:10.1103/PhysRevB.60.7764.

Rignot, E., S. Jacobs, J. Mouginot, and B. Scheuchl (2013), Ice-shelf melting around Antarctica, Science, 341(6143), 266–270, doi:10.1126/science.1235798.

Schaffer, J., R. Timmermann, J.E. Arndt, S. S. Kristensen, C. Mayer, M. Morlighem and D. Steinhage (2016), A global, high-resolution data set of ice sheet topography, cavity geometry, and ocean bathymetry, Earth System Science Data, 8, 543-557, doi: 10.5194/essd-8-543-2016.

Schmidtko, S., K. J., Heywood, A. F. Thompson, and S. Aoki (2014), Multidecadal warming of Antarctic waters, Science, 346, 1227–1231, doi:10.1126/science.1256117.

Shepherd, A. et al. (2018), Mass balance of the Antarctic Ice Sheet from 1992 to 2017, Nature, 558, 219-222, doi:10.1098/rsta.2006.1792.

Steig, E. J., Q. Ding, D. S. Battisti, and A. Jenkins (2012), Tropical forcing of Circumpolar Deep Water Inflow and outlet glacier thinning in the Amundsen Sea Embayment, West Antarctica, Annals of Glaciology, 53, 19–28, doi:10.3189/2012AoG60A110.

Thoma, M., A. Jenkins, D. Holland, and S. Jacobs (2008), Modelling Circumpolar Deep Water intrusions on the Amundsen Sea continental shelf, Antarctica, Geophysical Research Letters, 35, L18602, doi:10.1029/2008GL034939.

Turner, J., A. Orr, G. H. Gudmundsson, A. Jenkins, R. G. Bingham, C-D Hillenbrand, and T. J. Bracegirdle (2017), Atmosphere-ocean-ice interactions in the Amundsen Sea Embayment, West Antarctica, Reviews of Geophysics, 55, 235-276, doi:10.1002/2016RG000532.

Vaughan, D.G., J. C. Comiso, I. Allison, J. Carrasco, G. Kaser, R. Kwok, P. Mote, T. Murray, F. Paul, J. Ren, E. Rignot, O. Solomina, K. Steffen, and T. Zhang (2013), Observations: Cryosphere, in: Climate Change 2013: The Physical Science Basis. Contribution of Working Group I to the Fifth Assessment Report of the Intergovernmental Panel on Climate Change, T. F. Stocker, D. Qin, G.-K Plattner, M. Tignor, S. K. Allen, J. Boschung, A. Nauels, Y. Xia, V. Bex, and P. M. Midgley (eds.), Cambridge University Press, Cambridge, United Kingdom and New York, NY, USA, 65pp.

Walker, C. C., and A.S. Gardner (2017), Rapid drawdown of Antarctica's Wordie Ice Shelf glaciers in response to ENSO/Southern Annular Mode-driven warming in the Southern Ocean, Earth and Planetary Science Letters, 476, 100–110, doi:10.1016/j.epsl.2017.08.005.

Webber, B. G. M., K. J., Haywood, D. P. Stevens, P. Dutrieux, E. P. Abrahamsen, A. Jenkins, S. S. Jacobs, H. K. Ha, S. H. Lee, and T. W. Kim (2017) Mechanisms driving variability in the ocean forcing of Pine Island Glacier, Nature Communications, 8(14507), doi:10.1038/ncomms14507.

Weertman, J. (1974), Stability of the junction of an ice sheet and an ice shelf, Journal of Glaciology, 13(67), 3–11.

Wouters, B., A. Martin-Español, V. Helm, T. Flament, J. M. van Wessem, S. R. M. Ligtenberg, M. R. van den Broeke, and J. L. Bamber (2015), Dynamic thinning of glaciers on the Southern Antarctic Peninsula, Science, 348, 899–903, doi:10.1126/science.aaa5727.

Zhang, X., A. F. Thompson, M. M. Flexas, F. Roquet, and H. Bornemann (2016), Circulation and meltwater distribution in the Bellingshausen Sea: From shelf break to coast, Geophysical Research Letters, 43, 6402–6409, doi:10.1002/2016GL068998, 2016.

---

## Author Comment (AC3) · 21 Jun 2019

Our reply is written in blue.

**1 Summary**

This paper is a direct response to one of the geoengineering solutions presented in Moore et al 2018. The authors use a global ocean model to determine how an engineered submarine wall in the Amundsen Sea might affect the integrated basal melt rate of Antarctic ice shelves. They find that, as deep warm water is prevented from entering the ice shelf cavities of the Amundsen Sea, basal melt rates in other locations are increased. The net effect of the submarine wall is an integrated 10% decrease in basal melt from a control case.

**2 Recommendation**

This paper is timely and well-written. It provides a concise response to an earlier publication (Moore et al 2018), and illustrates one specific aspect of the complexity of geoengineering problems. Namely, that if heat is blocked from melting certain ice shelves, it may well go elsewhere and cause additional problems. There are many ways this paper could be expanded, however, the authors have done well in isolating and investigating a specific problem and containing it to a brief comment. I recommend publication after a few minor modifications.

Thank you very much for your encouraging comments. We have inspected if heat is accumulated. However we do not see a clear signal which we can be easily link to erected wall and described easily within the short "Brief communication" format. We detect in the Pacific Sector a distinct warming approximately along and slightly south of the Antarctic Circumpolar Current (ACC) and upstream (relative to the direction of the Antarctic Coastal Current) of the walled region at 700 m depth, for instance. Between the coastal current and the ACC, we also detect "filament-like" colder (compared to the control run) water spreading paths originating apparently from the Ross Ice Shelf region. We interpret this pattern as a modification of the circulation pattern.

South of 65°S, we detect a warming in the upper 100 meters of water column, but we do not consider this change as significant, because the atmospheric fluxes are fixed between both simulations. Any change in the ocean surface condition that does not influence the atmosphere, which in turn would modify the ocean forcing. This missing feedback can lead to wrong conclusions (Mikolajewicz and Maier-Raimer, 1994). Since we aim for a coupled atmosphere-ocean model that includes explicitly the ocean-ice shelf interaction, we have decided to analyze these features in more detail in a coupled atmosphere-ocean model system, which deems to be more appropriate.

Anyhow we are confident that our main results are robust: a) A wall shielding the Amundsen Sea Embayment reduces basal melting rates within the protected region, b) the rejected warm water masses flows along the wall westward, c) west of the wall warmer water masses drive enhanced basal melting.

**3 General comments**

I strongly encourage the authors to make the data used for this paper freely available online. Ideally, this would include model source code, forcing files, and code used for analysis. At the least, the authors should deposit a subset of model output used for the calculations in a repository such as Zenodo.

We followed your suggestion and changed the former "Data availability" section into "Code and Data availability":
The FESOM1.4 model code is available at https://swrepo1.awi.de/projects/fesom/ after registration. The here used atmospheric forcing data set named "CORE-II" (Large and Yeager, 2008) is freely accessible online (for example at https://data1.gfdl.noaa.gov/nomads/forms/core/COREv2.html). The topography data set RTOPO could be obtained from https://doi.pangaea.de/10.1594/PANGAEA.741917. The temporal average of the fractional basal melting changes between the CTRL and the WALL simulations is obtainable from Zenodo via https://dx.doi.org/10.5281/zenodo.3240250. The remaining data is available from the first author ÖG upon reasonable request.

There are several interesting points of discussion the authors do not address. As this paper is a direct comment on Moore et al 2018, it is up to the authors if they want to include further speculation. Here I list several of these points, for both the authors and the broader community to consider:

- Where does the heat go? If the integrated basal melting loss around Antarctica is 10% lower in the WALL experiment, the heat that would have caused melt must be somewhere else. Is it still in the ocean (and what part), transferred to the atmosphere or to sea ice melt, or somewhere else?

  As described above, we have analyzed if a warm water pols it created. However the vague signal does not allow to draw a strong conclusion, because the atmospheric flux are described and, hence, identical between our two sets of simulations. We deem it more appropriate to perform such a study with a model version under development, where the ocean and atmosphere is coupled. In this model the heat flux between atmosphere and ocean evolves freely and would allow quantifying the impact of heat flux changes.

- The model uses forcing for present day conditions. Under future warming scenarios, would the WALL experiment cause more or less integrated basal melt?

  We've planned the suggested experiments in a model version, where the ocean is interactively coupled to the atmosphere.

- Both this paper and Moore et al 2018 focus on sea level rise and investigate geoengineering possibilities that may also affect aspects of the climate outside of sea level. In this case of a wall, oceanic heat is redistributed and water circulation in ice shelf cavities in the Amundsen Sea is reduced. This has implications for the local ecosystem that is controlled by ice shelf cavity circulation (e.g., St. Laurent et al 2017). It would be useful to note that geoengineered 'solutions' to sea level rise affect more than just the target area

or problem, as the climate system is not separable into individual pieces.

Good point. We have missed this specific side effect. We follow your advice and add the following sentence:
"Iron is a micronutrient essential for algal production in the Amundsen Sea (St-Laurent et al., 2017) and the erected wall affects its availability. The wall blocks in inflow of warm and iron-rich CDW and influences the outflow of iron-rich glacial melt water coming from melting ice shelves. How the changed nutrient supply impacts the marine biological web or the uptake and sequestration of carbon dioxide by the ocean is unclear and goes beyond this study."

**4 Line comments**

Line 58-60 - It would be useful here to know that the WALL in your simulations does not match the wall in Moore et al 2018. Option to include a brief discussion why, or refer the reader to a later section.

We added the following new lines and adjusted the related discussion:
"The wall proposed by Moore et al. (2018), which blocks only the circulation in troughs leading directly to Pine Island and Thwaites Glaciers, would have a length of about 50—100 km and would need 10—50 km3 of material. By comparison, the construction of the Suez Channel required the excavation of about 1 km2 of material (Moore et al., 2018). The simulated wall (length of about 800 km) is substantially larger than the originally proposed wall in size and it shields the entire Amundsen Sea Embayment."

Line 81 - Remove the fraction; use only the 85% value

Done.

Line 135+ - A portion of the references are redundant.

We have checked and cleaned the references.

**5 References**

Large, W. G. and Yeager, S. G.: The global climatology of an interannually varying air–sea flux data set, Clim. Dyn., 33(2–3), 341–364, doi:10.1007/s00382-008-0441-3, 2008.
Mikolajewicz, U. and Maier-Reimer, E.: Mixed boundary conditions in ocean general circulation models and their influence on the stability of the model's conveyor belt, J. Geophys. Res., 99(C11), 22633–22644, doi:10.1029/94JC01989, 1994.

Moore, J. C., Gladstone, R., Zwinger, T., Wolovick, M. (2018). Geoengineer polar glaciers to slow sea-level rise. Nature, 555(7696), 303-305.

St-Laurent, P., P. L. Yager, R. M. Sherrell, S. E. Stammerjohn, and M. S. Dinniman (2017), Pathways and supply of dissolved iron in the Amundsen Sea (Antarctica), J. Geophys. Res. Oceans, 122, 7135–7162, doi:10.1002/2017JC013162.

---

## Author Response (AR2)

This document contains two sections.

The first three sections contain the replies. The last section highlights the changes of the revised article in comparison to the originally submitted article.

I would like to thank Mike Wolovik and the two anonymous reviewers for there engagement and open discussion. It has improved the manuscript.

| PDF page counter | Content |
| --- | --- |
| Page 2-5 | Replies (minor revision) |
| Page 6-19 | Markup version of revised article |

Editor Decision: Publish subject to minor revisions (review by editor) (21 Jul 2019) by Benjamin Smith

Comments to the Author:
Thanks for thorough responses to the referees. There are some places where the language in the MS is a little rough--

Thanks for taking the time to review the manuscript and for the valuable suggestions. Our comments and replies are in blue.

Lines 75-78: Not sore what "for instance" is meant to indicate. This phrase needs a prior general phrase so that the reader can tell what general condition the instance is of. I don't see that here.

We have reduced the unnecessary wordiness.

134: another "for instance" that isn't quite clear-- the previous list of parameters seems fairly comprehensive, so it's not like those are only a few of the parameters that might be relevant.

Yes we agree that it is a quite comprehensive list, but some may argue that also the ice properties, such as temperature and concentration of impurities, matter. Hence we have rephrased the entire sentence: "However, it is an open question if this triggers Marine Ice Sheet Instability in the other shelves, because the stability depends **strongly** on the distribution of pinning points, sloping of the bed, the depth and width of submarine troughs, and the softness of the bed, for instance."

150: "a by expert judgement merged data products" -- this seems to be garbled somehow

We've shortened this part and keep "data products"

167: move ',for instance,' directly after the word 'wall'

We have moved as suggested.

174: "may shifts also" - should be "may also shift'

Thanks for indicating this mistake, which we have corrected.

This is likely a partial list. Please look over the MS again to check for similar problems

Beside the above listed items, we have also changed the following.

Line 17 : Delete comma:
"coastal societies**,** and economic activities" =>
"coastal societies and economic activities"

Line 20 : Replace : "ground water" => "groundwater"

Line 23 : Replace : "modelled"  by "modeled"

Line 35 : Add comma:
"During its transport onto the continental shelf the water" =>

"During its transport onto the continental shelf**,** the water"

Line 49 : Replace "ice berg" => "iceberg"

Line 53/54 : Add comma :
"In addition to the erection of a submarine wall they" =>
"In addition to the erection of a submarine wall**,** they"

Line 57 : Add comma :
"In this paper we investigate how" =>
"In this paper**,** we investigate how"

Line 59 : Spelling: "sea level raise" => "sea level rise"

Line 71 : Add comma :
"ice shelf geometry and grounding line" =>
"ice shelf geometry**,** and grounding line".

Line 81 : Shortened "provide confirmation that the" => "confirm that the"

Line 84 : Delete comma :
"continental shelf break**,** and" =>
"continental shelf break and"

Line 88/89 : Rephrase :
"substantially larger than the originally proposed" =>
"substantially longer than the initially proposed"

Line 90 : Word order :
"experiment simulates accurately the ingress" =>
"experiment accurately simulates the ingress"

Line 102 : Delete comma :
"around Siple Island**,** because the" =>

"around Siple Island because the"

Line 105 : Add word :
"temperature reduces melting of" =>
"temperature reduces **the** melting of"

Line 105 : Shortened : "(Figure 2a and Figure 2c)." => "(Figure 2a and 2c)."

Line 113 : Add word :
"underneath Abbot" =>
"underneath **the** Abbot"

Line 115 : Rephrase :
"with the exception of the Amery Ice Shelf" => "except for the Amery Ice Shelf"

Line 117 : Add word :
"All above reported" =>
"All **the** above reported"

Line 143/144 : Word order :
"such as those seen in in Prydz Bay in front of Amery Ice Shelf or in the George VI Ice Shelf," =>
"such as those seen in the George VI Ice Shelf or in Prydz Bay in front of Amery Ice Shelf,"

Line 150-152 : Rephrase : "Therefore, we use RTOPO instead of most updated products. Since our

simulations are consistent with former studies using RTOPO, the quality of our simulations could be judged in the light of former studies." =>
"Therefore, we use RTOPO instead of **the** most updated products. Since our simulations are consistent with **prior** studies using RTOPO, the quality of our simulations could be judged in the light of **previous** studies."

Line 153 : Add word :
"Shelf into walled" =>
"Shelf into **the** walled"

Line 154 : Change word :
"ice was grounded" =>
"ice **were** grounded"

Line 155 : Add comma :
"However we would" =>
"However**,** we would"

Line 157-159 : Rephrase : "These simulations would also uncover if enhanced melting at the western end of the wall may open a backdoor that open a second route" =>
"These simulations **could** also uncover if enhanced melting at the western end of the wall **unlocks** a backdoor that open**s** a second route"

Line 161/162 : Correct :
"warm water masses flow**s**" =>
"warm water masses flow"

Line 166 : Change word :
"larger than the originally proposed" =>
"larger than the **initially** proposed"

Line 168/169: Rephrase : "Our results suggest that a too small wall blocking only the water flow in the troughs leading to Pine Island, for instance, might" =>
"Our results suggest that a too-small wall**, for instance,** blocking only the water flow in the troughs leading to Pine Island might"

Line 170 : Delete comma :
"hemisphere)**,** if it is not hindered" =>
"hemisphere) if it is not hindered"

Line 172 : Add comma : "However**,** a small"

Line 175 : Rephrase :
"global warming (Miller et al., 2006) may shifts also the coast easterly" =>
"global warming (Miller et al., 2006) **might also shift** the coast easterly"

Line 181 : Add word :
"The wall blocks in inflow" =>
"The wall blocks in **the** inflow"

Line 203/204 : Rephrase :

[revised manuscript text omitted]